# Global, regional, and national burden of early-onset OA attributable to high BMI: 1990–2021 estimates and 2036 projections from the global burden of disease study

**Binbin Zhang☯, Bin Dou☯, Wenzuo Gu, Chuan Lu, Qi Yan, Dawa Zhaxi, Yu Wang, Jiale Xu, Kewen Li**  *

Qinghai University Affiliated Hospital (The Clinical Medical School), Qinghai University, Xining, Qinghai, China

☯ These authors contributed equally to this work and share first authorship.
* qdfykwl@126.com

## Abstract

### Objective

High BMI is a critical risk factor for early-onset OA (diagnosed before age 55). This study aimed to analyze global trends in the age-standardized disability-adjusted life years rates (ASDR) attributable to high BMI from 1990 to 2021.

### Methods

Data from the Global Burden of Disease 2021 (GBD 2021) study were analyzed to assess early-onset OA attributable to high BMI across 204 countries, 21 GBD regions, and 5 Socio-Demographic Index (SDI) tiers. Temporal trends in ASDR were quantified using the estimated annual percentage change (EAPC) and Joinpoint regression. Age-period-cohort models and decomposition analysis identified drivers of burden, while inequality was assessed using the Slope Index of Inequality (SII) and Concentration Index (CI). ARIMA models projected trends to 2036.

### Results

The disease burden of early-onset OA attributable to high BMI increased significantly between 1990 and 2021. The DALYs for early-onset knee and hip OA rose by 203.01% and 170.12%, respectively, with average annual percentage changes (AAPC) of 1.39% and 1.40%. The age-period-cohort analysis indicated that DALYs risk increased with age and period, and later birth cohorts experienced a higher burden of early-onset OA attributable to high BMI. Decomposition analysis revealed that population growth was the primary driver of the rising disease burden. There were significant absolute and relative inequalities in the burden of early-onset OA attributable to high BMI, as measured by the SDI. Countries with higher SDI bore a greater

**Data availability statement:** All relevant data are within the manuscript and its Supporting Information files.

**Funding:** This study was funded by 2024 Provincial and ministerial joint construction of Central Asia high disease causes and prevention of the national key laboratory-Qinghai Workstation Joint Fund (SKL-HIDCA-2024-QH1) The funders had no role in study design, data collection and analysis, decision to publish, or preparation of the manuscript.

**Competing interests:** The authors have declared that no competing interests exist.

burden. The SII demonstrated that the ASDR gap between countries with the highest and lowest SDI widened steadily from 1990 to 2021. Predictive analysis suggested that the burden of early-onset OA attributable to high BMI will continue to increase over the next 15 years.

## Conclusion

From 1990 to 2021, the global burden of early-onset OA attributable to high BMI showed a consistent upward trend, with significant inequalities across countries. The disease burden is projected to grow further in the future.

## 1 Introduction

Osteoarthritis (OA), a common degenerative joint disease, is initially characterized by joint pain, swelling, and restricted mobility [1]. As the disease progresses, joint deformities may develop, and in severe cases, paralysis can occur. OA has emerged as a significant global health challenge. Recent studies have shown a steady increase in its incidence and prevalence worldwide, with 527,811,871 cases reported in 2019 [2]. This rising trend imposes substantial economic and healthcare burdens on many countries [3,4]. Traditionally, OA has been considered an age-related disease [5]. However, recent years have seen an increase in early-onset OA (primary OA diagnosed before the age of 55) among younger and middle-aged individuals due to factors such as unhealthy lifestyles and sports injuries [6]. An 18-year analysis of 90,897 OA patients from seven U.S. hospitals showed that the proportion of younger patients (aged 18–45) rose from 6.2% in 2001 to 22.7% in 2019. During this period, the number of patients with comorbid anxiety and depression also increased [7]. These findings indicate that OA is no longer limited to the elderly population. The increasing prevalence of early-onset OA among younger individuals highlights the urgent need for enhanced awareness and proactive intervention.

In recent years, the global obesity rate has risen alarmingly [8]. Influenced by lifestyle changes, including the prevalence of high-calorie diets, reduced physical activity, and increased sedentary behavior, an increasing number of young individuals are becoming obese [9]. Obesity is one of the key modifiable risk factors for early-_onset OA, along with trauma, developmental abnormalities, and excessive joint use [10,11]. The potential association between obesity and early-onset OA has attracted increasing attention in recent research. A study across all age groups showed that the age-standardized disability-adjusted life years rate (ASDR) caused by OA attributed to high body mass index (BMI) increased from 21.24 per 100,000 population in 1990 to 31.9 in 2019, representing an approximate 50.18% increase [12]. This trend indicates that obesity not only exacerbates the burden of OA but also significantly impacts the productivity of young and middle-aged individuals. Such effects may have far-reaching adverse implications for societal development. Research on the disease burden of OA has made notable progress. Some studies have focused on specific regions or countries, providing detailed analyses of local OA

burdens [13,14]. In contrast, others have adopted a global perspective, offering comprehensive assessments of skeletal diseases or OA burdens at different anatomical sites [15–17]. Although these studies have contributed to the analysis of OA disease burden, there are still some gaps and deficiencies in the current study. First, there is a lack of systematic evaluations addressing the impact of high BMI on the disease burden of early-onset OA. Second, existing studies generally fail to conduct in-depth analyses of temporal trends, leaving the future burden of early-onset OA attributable to high BMI largely unexplored. These limitations highlight the need for further investigation into the long-term effects of high BMI on early-onset OA and the dynamic changes in its disease burden. Such research is essential for informing more comprehensive prevention and intervention strategies.

OA, a chronic progressive disease, involves irreversible structural changes. Therefore, proactive management measures are crucial in the early stages of the disease. The global obesity rate among young people is continuously increasing, and obesity is a significant risk factor for the onset and progression of OA, potentially driving the incidence of early-onset OA. In this context, systematically analyzing the disease burden of early-onset OA caused by obesity in different countries is highly important. Significant differences in public health awareness and healthcare system performance exist across countries with varying levels of economic and social development, particularly in less developed countries, where these disparities hinder the effective management and prevention of obesity and early-onset OA. To deepen the understanding of the epidemiological characteristics of early-onset OA caused by obesity, this study uses data from the 2021 Global Burden of Disease (GBD) study [18] to systematically explore the global disease burden of early-onset OA attributed to high BMI. The specific objectives include: (1) Describing the trends in DALYs and ASDR for early-onset OA caused by high BMI in men and women from 1990 to 2021 at the global, regional (including 5 Socio-Demographic Index (SDI) regions and 21 GBD regions), and national (204 countries) levels. (2) Exploring the association between SDI levels and ASDR across regions and countries through health inequality analysis. (3) Analyzing the patterns of DALYs changes worldwide from the perspectives of age, period, and cohort effects. (4) Decomposing the DALYs composition through analysis of aging, population growth, and epidemiological changes. (5) Using the ARIMA model to predict the disease burden of early-onset OA caused by high BMI for the next 15 years based on data from 1990 to 2021.

## 2 Methods

### 2.1 Data sources

The data utilized in this study were sourced from the Global Burden of Disease 2021 (GBD 2021) database, officially released by the Institute for Health Metrics and Evaluation (IHME) at the University of Washington [18]. This database provides comprehensive disease burden data for 371 diseases and injuries across 204 countries and regions, spanning the years 1990–2021. The data can be accessed via the following link: https://ghdx.healthdata.org/gbd-2021. Detailed data screening criteria are outlined in Appendix 1. The settings for data retrieval are as follows: Estimate (Risk factors), measures (DALYs), Metric (rate and percentage), risk factors (high BMI), cause (knee OA and hip OA), location (global, 5 SDI regions, 21 GBD regions, and 204 countries and regions), age (30–34, 35–39, 40–44, 45–49, 50–54), sex (both, male, and female), year (1990–2021). After the retrieval is completed, the data will be downloaded as a.csv format file and sent to a personal email for subsequent analysis.

### 2.2 Measures and definitions of disease burden

The GBD 2021 study has thoroughly outlined the methods for estimating disease burdens attributable to various risk factors [19]. In this study, high BMI is defined as BMI ≥ 25 kg/m$^2$, encompassing both overweight and obesity.

DALYs serve as a comprehensive metric to evaluate health loss from disease onset to death. DALYs comprise the sum of years of life lost (YLL) and years lived with disability (YLD). In the GBD 2021, no mortality data are available for early-onset OA attributable to high BMI; therefore, YLD and DALY estimates are identical. Given that OA does not directly cause mortality, DALYs were selected as the core metric to assess the disease burden in this study.

To ensure comparability across populations and minimize variations due to age structure differences, ASDR were calculated based on the world standard population age structure defined in the GBD 2021 report. ASDR, expressed per 100,000 population, provides a standardized measure of the overall health impact of early-onset OA attributable to high BMI. This metric is widely used for comparing disease burdens across regions. The formula for calculating ASDR for early-onset OA caused by high BMI is outlined below [20]:

$$Age-standardized/: rate = \frac{\sum_{i=1}^{A} \alpha_i w_i}{\sum_{i=1}^{A} w_i} \times 100\ 000$$

$\alpha_i$ being the age-specific rate in the $i$-th age group, $w_i$ being the number of people in the $i$-th corresponding age group among the standard population and $A$ being the number of age groups.

The Socio-demographic Index (SDI) is a composite metric designed to assess development levels closely linked to health outcomes [21]. It is based on three core variables: per capita lag-distributed income, average educational attainment among individuals aged 15 years and older, and fertility rates among women under 25 years of age. The SDI ranges from 0 to 1, reflecting the health-related development levels of a region. An SDI value of 0 indicates the lowest theoretical level of development, while a value of 1 represents the highest theoretical level. Based on SDI thresholds, the 204 countries and territories are categorized into five development levels: low SDI regions (0–0.4658), low-middle SDI regions (0.4658–0.6188), middle SDI regions (0.6188–0.7120), high-middle SDI regions (0.7120–0.8103), and high SDI regions (0.8103–1.0000). This classification facilitates a more detailed examination of health conditions and associated risk factors across regions with varying levels of development.

## 2.3  Descriptive analysis

This study investigates ASDR values attributable to high BMI induced early-onset OA across three levels: global, regional (including 5 SDI regions and 21 GBD geographic regions), and national (204 countries and territories) for the years 1990 and 2021. Furthermore, it examines gender differences in ASDR values to provide a comprehensive understanding of disparities in disease burden.

## 2.4  Time trend analysis

To reflect the change of ASDR in 2021 compared to 1990, this study utilizes the estimated annual percentage change (EAPC) to quantify the overall trend in the burden of early-onset OA attributable to high BMI. An EAPC greater than 0 indicates an increasing trend in the research indicator, while an EAPC less than 0 indicates a decreasing trend [22]. In this equation, $\beta$ represents the regression coefficient. The formula for calculating EAPC (%) is as follows:

$$EAPC(\%) = 100 \times (exp(\beta) - 1)$$

This study employs Joinpoint Regression Analysis to investigate the overall trend and local changes in the ASDR of early-onset OA attributable to high BMI. A log-linear regression model was used to calculate the Average Annual Percentage Change (AAPC) and Annual Percentage Change (APC). The ASDR of early-onset OA attributable to high BMI were set as the dependent variable, while year was treated as the independent variable. In the model settings, the heteroscedasticity error option was assumed to be constant variance, and the maximum number of joinpoints was limited to five. The log-linear model (lny = xb) was chosen, and the Bayesian Information Criterion (BIC) was applied to determine the optimal number of change points [23–24]. The trend was classified as follows: if the 95% confidence interval (CI) of either the APC or AAPC is entirely above zero, the trend is considered significantly increasing; if both are below zero, the trend is significantly decreasing; and if the CI includes zero, the trend is not statistically significant. A two-tailed test was

performed, with the significance level set at α = 0.05. The specific formulas for calculating the APC and AAPC for each phase are as follows:

$$APC = [\exp(\beta) - 1] \times 100$$

$$AAPC = \left[\exp\left(\frac{\sum w_i b_i}{\sum w_i}\right) - 1\right] \times 100$$

$b_i$ is the slope coefficient of the $i$-th segment. Here, $i$ is used to index each segment within the required year range. The width $w$ of the segment interval weights the regression coefficients of each interval. $w_i$ represents the interval width of each piecewise function (that is, the number of years included in the interval), and $\beta$ is the regression coefficient corresponding to each interval.

## 2.5 Age-period-cohort analysis

The Age - Period - Cohort (APC) model is a statistical tool used to analyze the hanges in disease incidence or mortality over time, and it is widely applied in epidemiological research [25–27]. This model is based on three times dimensions: age, period, and birth cohort. The age effect reflects the different susceptibilities of individuals to diseases at different physiological age stages. The period effect reveals the impact of factors such as society, economy, environment, and medical technology on disease rates or mortality rates within a specific time. The cohort effect reflects the influence of diseases on people born in the same time period due to their common early – life experiences and changes in the social environment. According to the data requirements of the Age – Period – Cohort model, in this study, the age range was divided into 30–54 years old, with each group consisting of 5 – year intervals, resulting in a total of 5 age groups. Since the time span from 1990 to 2021 could not be evenly divided into 5 – year intervals, this study selected the period from 1992 to 2021 as the analysis interval and divided it into 6 consecutive 5 – year time periods: 1992–1996 (Period 1994), 1997–2001 (Period 1999), 2002–2006 (Period 2004), 2007–2011 (Period 2009), 2012–2016 (Period 2014), and 2017–2021 (Period 2019). The birth cohorts were calculated by subtracting the age from the period, resulting in a total of 10 cohorts (e.g., 1938–1946, 1943–1951... 1983–1991). In this study, a general linear model was used to evaluate the interaction between the period effect and the birth cohort effect, and the rate ratio (RR) was calculated. The specific estimation function was statistically tested using the Wald chi – square test, and the two – sided test level was set at α = 0.05. The basic formula is as follows:

$$(R_{ijk}) = \ln\left(\frac{y_{ij}}{n_{ij}}\right) \mu + \alpha_i + \beta_j + \gamma_k + \varepsilon$$

$R_{ijk}$ represents the DALYs rate of the $k$-th birth cohort in the $i$-th age group during the $j$-th period. $y_{ij}$ represents the number of DALYs, $n_{ij}$ represents the number of exposed populations in the same period, $\mu$ represents the intercept of the regression equation, and $\varepsilon$ represents the random error that follows a normal distribution. Here, $i$, $j$, $k$ and represent the age, period, and birth cohort groups respectively; $\alpha_i$, $\beta_j$, and $\gamma_k$ represent the age effect, period effect, and birth cohort effect respectively.

## 2.6 Decomposition analysis

Decomposition analysis is a data analysis method. Its objective is to decompose the changes in the disease burden into three key factors: population, aging, and epidemiological changes, so as to clarify the contribution degree of each factor to the overall changes [28]. In this study, decomposition analysis was conducted to deeply analyze the changes in the disease burden of early-onset OA attributable to high BMI from 1990 to 2021.

## 2.7 Health inequality analysis

This study employs the Slope Index of Inequality (SII) and the Concentration Index (CI) to evaluate disparities in the ASDR for early-onset OA attributable to high BMI across different countries, providing a comprehensive analysis of health inequalities. The SII is a critical measure of absolute inequality, reflecting the difference in health outcomes between the highest and lowest socioeconomic groups. Derived through regression analysis, the horizontal axis represents the cumulative percentage of the population, ranked by socioeconomic indicators (e.g., per capita GDP), ranging from 0 to 1. The midpoint of each subgroup's horizontal coordinate corresponds to its relative socioeconomic status, while the vertical axis represents the ASDR values. A higher SII indicates a greater degree of absolute health inequality [29]. The CI, on the other hand, is a measure of relative inequality calculated based on the concentration curve. On this curve, the horizontal axis represents the cumulative percentage of the population ranked by socioeconomic indicators, while the vertical axis represents the cumulative percentage of the health indicator (ASDR). The CI is defined as the area between the concentration curve and the diagonal line (representing perfect equality) and ranges from −1–1. A negative CI indicates that the health indicator is more concentrated among populations with lower socioeconomic status, while a positive CI suggests that it is more concentrated among those with higher socioeconomic status [30].

## 2.8 Forecast analysis

The ARIMA (Autoregressive Integrated Moving Average) model is a widely used model in time series analysis. It is composed of three parts: autoregressive (AR), integration (I), and moving average (MA) [31]. The basic form of the model is ARIMA (p, d, q), where p represents the order of autoregression, d represents the order of differencing, and q represents the order of moving average [32]. The ARIMA model can effectively predict future data by combining the autoregressive and moving average characteristics of time series. In this study, the ARIMA model was used to predict the disease burden of early-onset OA caused by high BMI in the next 15 years. In terms of model parameter selection, we compared the goodness-of-fit of different models according to the Akaike Information Criterion (AIC) and the Bayesian Information Criterion (BIC), and selected the model with the smallest criterion value as the optimal model. After the model was established, the Ljung-Box Q test was used to conduct a white noise hypothesis test on the residuals. When the P-value of the test is greater than 0.05, it indicates that the model residuals conform to the white noise hypothesis, which means that the information in the time series has been fully extracted. If the P-value is less than or equal to 0.05, it indicates that the model needs to be further adjusted to improve its prediction performance.

## 3 Result

Globally, the ASDR of early-onset knee OA attributable to high BMI increased from 26.78 cases per 100,000 population (95% UI, −2.40–74.00) in 1990 to 41.01 cases per 100,000 population (95% UI, −4.05–110.33) in 2021 (Table 1). Similarly, the ASDR of early-onset hip OA attributable to high BMI rose from 2.58 cases per 100,000 population (95% UI, −0.21–7.04) in 1990 to 3.62 cases per 100,000 population (95% UI, −0.33–9.96) in 2021 (Table 1). Based on the estimated annual percentage change (EAPC), the ASDR of early-onset knee OA (EAPC = 1.56; 95% CI, 1.49–1.64) and early-onset hip OA (EAPC = 1.20; 95% CI, 1.16–1.23) attributable to high BMI both demonstrated an increasing trend between 1990 and 2021 (Table 1, Fig 1).When analyzed by sex, the ASDR for early-onset knee and hip OA attributable to high BMI showed an upward trend for both males and females over the same period (Table 1, Fig 1). Notably, the ASDR of early-onset knee OA was consistently higher in females than in males. Conversely, the ASDR of early-onset hip OA was generally higher in males than in females (Fig 2).

When categorized by SDI regions, the disease burden of early-onset knee OA attributable to high BMI was highest in high SDI regions. The ASDR increased from 36.32 cases per 100,000 population (95% UI, −3.51–99.65) in 1990 to 48.32 cases per 100,000 population (95% UI, −5.25–126.92) in 2021 (Table 1, Fig 1). According to the results of EAPC, all five SDI regions experienced an increase in ASDR from 1990 to 2021. Among these, low-middle SDI regions exhibited the

**Table 1. The ASDR of early-onset osteoarthritis attributed to high BMI in 1990 and 2021 for both sexes by SDI quintiles and by GBD regions, with EAPC from 1990 to 2021.**

| Location | Knee OA | | | Hip OA | | |
|---|---|---|---|---|---|---|
| | ASDR in 1990 (per 100,000) | ASDR in 2021 (per 100,000) | EAPC of ASDR (%) 1990–2021 | ASDR in 1990 (per 100,000) | ASDR in 2021 (per 100,000) | EAPC of ASDR (%) 1990–2021 |
| Global | 26.78 (−2.4,74) | 41.01 (−4.05,110.33) | 1.56 (1.49,1.64) | 2.57 (−0.21,7.04) | 3.62 (−0.33,9.96) | 1.2 (1.16,1.23) |
| **Sex** | | | | | | |
| Female | 32.85 (−2.9,90.5) | 50.02 (−4.97,134.26) | 1.56 (1.48,1.63) | 2.52 (−0.2,6.92) | 3.56 (−0.32,9.77) | 1.22 (1.19,1.25) |
| Male | 20.91 (−1.9,58.2) | 32.04 (−3.13,86.53) | 1.54 (1.47,1.6) | 2.62 (−0.22,7.23) | 3.67 (−0.34,10.14) | 1.18 (1.13,1.22) |
| **SDI regions** | | | | | | |
| High SDI | 36.32 (−3.51,99.65) | 48.32 (−5.25,126.92) | 0.97 (0.93,1.01) | 5.07 (−0.45,13.97) | 6.89 (−0.68,18.62) | 1.17 (1.1,1.24) |
| High-middle SDI | 28.95 (−2.65,80.56) | 46.39 (−4.62,124.09) | 1.76 (1.66,1.85) | 2.85 (−0.24,7.88) | 3.89 (−0.36,10.66) | 1.05 (1,1.1) |
| Middle SDI | 26.59 (−2.27,74.26) | 44.25 (−4.28,119.44) | 1.89 (1.79,1.99) | 1.63 (−0.13,4.53) | 2.94 (−0.26,8.14) | 1.98 (1.96,1.99) |
| Low-middle SDI | 17.46 (−1.52,48.81) | 31.31 (−2.96,85.85) | 2.07 (2,2.14) | 1.32 (−0.1,3.74) | 2.65 (−0.22,7.31) | 2.43 (2.38,2.49) |
| Low SDI | 14.92 (−1.15,42.54) | 24.22 (−2.07,67.8) | 1.66 (1.61,1.71) | 1.21 (−0.08,3.48) | 2.1 (−0.15,5.93) | 1.87 (1.81,1.92) |
| **GBD regions** | | | | | | |
| Andean Latin America | 39.57 (−3.97,105.71) | 55.14 (−6.3,142.02) | 1.14 (1.11,1.18) | 2.8 (−0.24,7.58) | 4.1 (−0.4,11.13) | 1.29 (1.26,1.32) |
| Australasia | 40.7 (−4.08,110.03) | 56.42 (−6.26,148.42) | 1.06 (1,1.11) | 5.41 (−0.5,14.91) | 8.6 (−0.86,23.7) | 1.54 (1.44,1.63) |
| Caribbean | 36.12 (−3.33,97.3) | 48.38 (−5.23,126.95) | 1 (0.95,1.04) | 2.71 (−0.22,7.57) | 3.69 (−0.35,10.11) | 1.1 (1.05,1.15) |
| Central Asia | 24.26 (−2.38,65.75) | 29.37 (−3.21,76.23) | 0.64 (0.63,0.65) | 3.6 (−0.32,9.98) | 4.56 (−0.44,12.53) | 0.81 (0.79,0.83) |
| Central Europe | 28.24 (−2.72,76.97) | 34.36 (−3.5,91.17) | 0.66 (0.64,0.67) | 3.94 (−0.33,10.91) | 5.15 (−0.47,13.84) | 0.88 (0.85,0.9) |
| Central Latin America | 40.98 (−4.16,108.76) | 54 (−6.19,137.68) | 0.89 (0.87,0.9) | 3.19 (−0.29,8.77) | 4.29 (−0.44,11.56) | 0.88 (0.84,0.93) |
| Central Sub-Saharan Africa | 15.1 (−1.16,43.61) | 27.08 (−2.24,75.97) | 1.88 (1.82,1.94) | 1.35 (−0.09,4.23) | 2.54 (−0.19,7.37) | 2.01 (1.93,2.09) |
| East Asia | 27.58 (−2.33,79.92) | 52.67 (−4.87,144.12) | 2.53 (2.33,2.72) | 1.1 (−0.08,3.22) | 2.38 (−0.19,6.69) | 2.63 (2.55,2.7) |
| Eastern Europe | 28.42 (−2.69,76.8) | 36.47 (−4.05,94.15) | 0.86 (0.84,0.88) | 3.94 (−0.31,11) | 5.39 (−0.5,14.76) | 1.09 (1.07,1.11) |
| Eastern Sub-Saharan Africa | 15.2 (−1.14,42.88) | 23.54 (−1.97,65.55) | 1.43 (1.41,1.46) | 1.43 (−0.09,4.17) | 2.37 (−0.17,6.77) | 1.68 (1.64,1.72) |
| High-income Asia Pacific | 28.53 (−2.43,82.36) | 37.38 (−3.44,105.62) | 0.95 (0.9,1) | 2.46 (−0.19,6.94) | 3.37 (−0.27,9.53) | 1.06 (1.01,1.11) |
| High-income North America | 49.05 (−5.08,130.8) | 59.64 (−7.06,152.27) | 0.68 (0.54,0.82) | 7.9 (−0.74,21.63) | 10.47 (−1.1,28.27) | 1.3 (1.17,1.42) |
| North Africa and Middle East | 33.72 (−3.43,90.42) | 49.01 (−5.83,124.73) | 1.21 (1.2,1.23) | 2.39 (−0.22,6.67) | 3.98 (−0.43,10.49) | 1.62 (1.6,1.65) |
| Oceania | 39.84 (−3.79,107.04) | 49.39 (−5.32,133.03) | 0.62 (0.54,0.71) | 2.47 (−0.2,6.94) | 2.96 (−0.28,8.3) | 0.47 (0.34,0.59) |
| South Asia | 13.6 (−1.06,39.17) | 27.5 (−2.31,78.39) | 2.61 (2.48,2.74) | 1.03 (−0.07,2.94) | 2.39 (−0.19,6.67) | 3.09 (2.97,3.21) |
| Southeast Asia | 16.08 (−1.24,46.44) | 28.05 (−2.45,78.09) | 1.89 (1.82,1.96) | 1.15 (−0.07,3.34) | 2.05 (−0.15,5.76) | 1.97 (1.91,2.04) |
| Southern Latin America | 40.93 (−4.25,110.27) | 54.61 (−6.23,136.8) | 0.95 (0.89,1.01) | 4.9 (−0.48,13.76) | 7.63 (−0.81,20.77) | 1.46 (1.34,1.57) |
| Southern Sub-Saharan Africa | 32.12 (−3.03,86.77) | 42.96 (−4.68,109.11) | 0.93 (0.9,0.96) | 3.71 (−0.3,10.33) | 5.28 (−0.48,14.51) | 1.14 (1.13,1.15) |
| Tropical Latin America | 35.57 (−3.38,96.23) | 47.94 (−5.22,125.66) | 0.97 (0.95,0.98) | 2.99 (−0.25,8.26) | 4.27 (−0.4,11.82) | 1.2 (1.18,1.22) |
| Western Europe | 31.63 (−2.99,86.48) | 40.06 (−4.1,107.5) | 0.76 (0.72,0.8) | 5.09 (−0.44,13.93) | 6.98 (−0.66,19.08) | 1.02 (0.94,1.09) |
| Western Sub-Saharan Africa | 22.51 (−1.89,63.57) | 33.98 (−3.22,91.84) | 1.32 (1.24,1.4) | 1.99 (−0.14,5.61) | 3.04 (−0.25,8.42) | 1.26 (1.19,1.33) |

most significant growth (EAPC = 2.07; 95% CI, 2.00–2.14) (Table 1, Fig 1). When analyzed by sex, males had the highest ASDR in high-SDI regions (41.64 cases per 100,000 population; 95% UI, −4.75–108.91). However, the largest increase in ASDR for males occurred in low-middle SDI regions (EAPC = 2.28; 95% CI, 2.21–2.34) (S1 Table, Fig 1). For females,

Early-onset knee osteoarthritis          Early-onset hip osteoarthritis

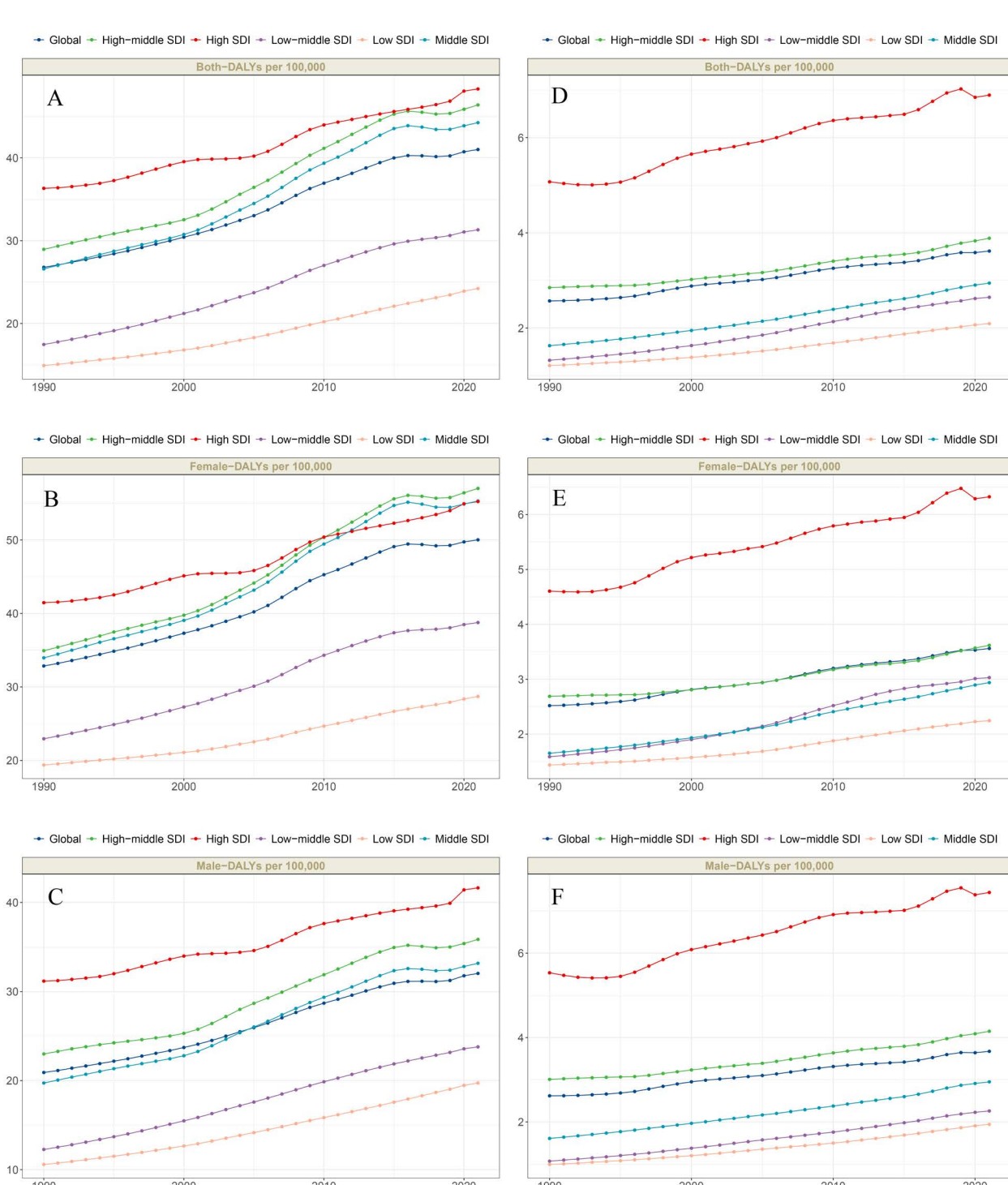

**Fig 1. The trends in ASDR of early-onset OA attributable to high BMI globally and across five SDI regions from 1990 to 2021.** Note: A-C, Early-onset knee OA attributable to high BMI; D-F, Early-onset hip OA attributable to high BMI. Abbreviations: SDI, Sociodemographic Index; ASDR, age-standardized disability-adjusted life years rate; BMI, Body mass index.

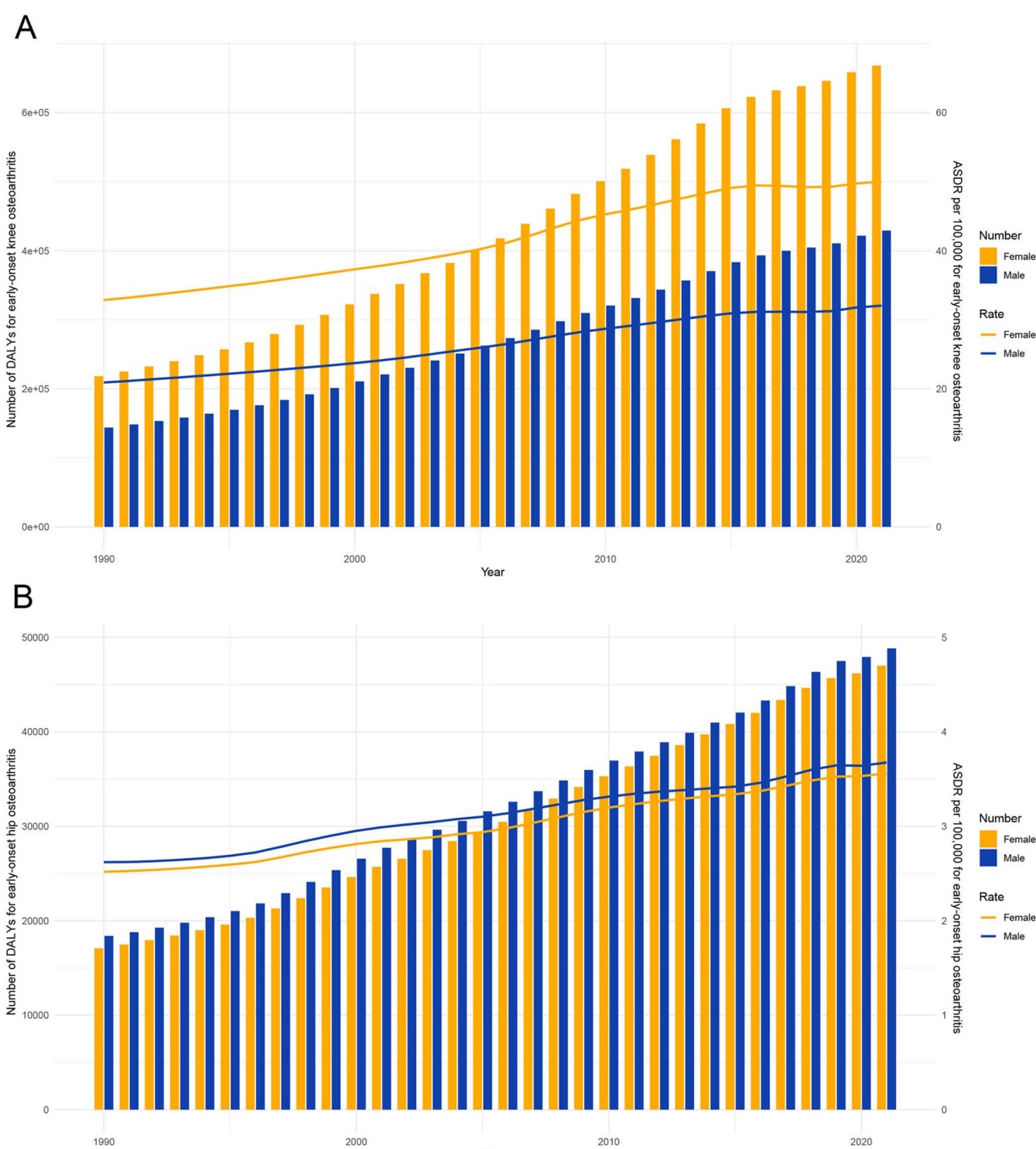

**Fig 2. Global burden of disease for early-onset OA attributable to high BMI by year and sex.** Note: A, Early-onset knee OA attributable to high BMI; B, Early-onset hip OA attributable to high BMI. Abbreviations: BMI, Body mass index.

the highest ASDR was observed in high-middle SDI regions (57.02 cases per 100,000 population; 95% UI, −5.75–152.08), while the most significant growth occurred in low-middle SDI regions (EAPC = 1.89; 95% CI, 1.81–1.98) (S2 Table, Fig 1).

Similarly, the disease burden of early-onset hip OA attributable to high BMI was also highest in high SDI regions. The ASDR increased from 5.07 cases per 100,000 population (95% UI, −0.45–13.97) in 1990 to 6.89 cases per 100,000 population (95% UI, −0.68–18.62) in 2021 (Table 1, Fig 1). According to the results of EAPC, all five SDI regions exhibited an increase in ASDR between 1990 and 2021. Among them, low-middle SDI regions showed the most significant growth (EAPC = 2.43; 95% CI, 2.38–2.49) (Table 1, Fig 1). When analyzed by sex, males exhibited the highest ASDR in high SDI regions (7.44 cases per 100,000 population; 95% UI, −0.76–20.04). However, the largest increase in ASDR for males occurred in low-middle SDI regions (EAPC = 6.32; 95% CI, −0.60–17.37) (S1 Table, Fig 1). For females, the highest ASDR was observed in high-middle SDI regions (57.02 cases per 100,000 population; 95% UI, −5.75–152.08), while the most significant growth occurred in low-middle SDI regions (EAPC = 2.35; 95% CI, 2.27–2.44) (S2 Table, Fig 1).

At the GBD region level, the disease burden of early-onset knee OA attributable to high BMI was highest in High-income North America. The ASDR increased from 49.05 cases per 100,000 population (95% UI, −5.08–130.80) in 1990 to 59.64 cases per 100,000 population (95% UI, −7.06–152.27) in 2021 (Table 1). According to the EAPC results, ASDR increased in all 21 regions from 1990 to 2021, with South Asia having the greatest increase in ASDR (EAPC = 2.61; 95% CI: 2.48–2.74) (Table 1). By sex, ASDR for males (52.65 cases per 100,000 population; 95% UI: −6.44–134.03) was highest in High-income North America. ASDR for females (9.52 cases per 100,000 population; 95% UI: −0.96–25.94) was highest in East Asia. According to the EAPC results, the increase in male ASDR was greatest in South Asia (EAPC = 2.76; 95% CI: 2.65–2.87, the increase in female ASDR was greatest in East Asia (EAPC = 3.16; 95% CI: 3.09–3.22) (S1-S2 Tables).

Early-onset hip OA attributed to high BMI had the highest ASDR (10.47 cases per 100,000 population; 95% UI: −1.1–28.27) in High-income North America (Table 1). According to the EAPC results, ASDR increased in all 21 regions in 2021 compared with 1990, with South Asia having the greatest increase in ASDR (EAPC = 3.09; 95% CI: 2.97–3.21) (Table 1). By sex, ASDR for both males (11.44 cases per 100,000 population; 95% UI: −1.23–30.9) and females (9.52 cases per 100,000 population; 95% UI: −0.96–25.94) was highest in High-income North America with the highest increase in South Asia (male EAPC result = 3.16; 95% CI: 3.09–3.22), (female EAPC result = 2.96; 95% CI: 2.79–3.13) (S1-S2 Tables).

At the national level, in 2021, the highest ASDR of early-onset knee OA attributable to high BMI was observed in the Cook Islands, with 85.73 cases per 100,000 population (95% UI, −11.97–211.54). According to the results of EAPC, ASDR increased in all countries between 1990 and 2021, with the largest growth recorded in Bangladesh (EAPC = 3.48; 95% CI, 3.33–3.63) (S3 Table, Fig 3). When analyzed by sex, both males (67.81 cases per 100,000 population; 95% UI, −8.27 to 165.46) and females (102.59 cases per 100,000 population; 95% UI, −14.23 to 253.09) showed the highest ASDR for early-onset knee OA attributable to high BMI in the Cook Islands. The most significant ASDR increase for males occurred in Botswana (EAPC = 3.71; 95% CI, 3.61–3.81) (S1 Table, S1 Fig), while for females, the greatest growth was observed in Bangladesh (EAPC = 3.35; 95% CI, 3.35–3.66) (S2 Table, S2 Fig).

For early-onset hip OA attributable to high BMI, Australia reported the highest ASDR in 2021, with 8.66 cases per 100,000 population (95% UI, −0.85–23.89). Based on the results of EAPC, ASDR showed an increasing trend in all countries between 1990 and 2021, with the largest growth observed in Bangladesh (EAPC = 4.11; 95% CI, 3.94–4.29) (S3 Table, Fig 3). For males, the highest ASDR in 2021 was recorded in Australia at 9.85 cases per 100,000 population (95% UI, −1.03–27.13). For females, the United States of America exhibited the highest ASDR, with 9.98 cases per 100,000 population (95% UI, −1.01–27.10). The greatest ASDR increase for males was observed in Botswana (EAPC = 4.09; 95% CI, 3.95–4.23) (S1 Table, S1 Fig), while for females, the largest growth occurred in Bangladesh (EAPC = 4.06; 95% CI, 3.90–4.22) (S2 Table, S2 Fig).

The results of the Joinpoint Regression Analysis are shown in Fig 4. For early-onset knee OA attributable to high BMI, although the ASDR generally exhibited an increasing trend (AAPC = 1.39%, 95% CI: 1.36%–1.43%), significant

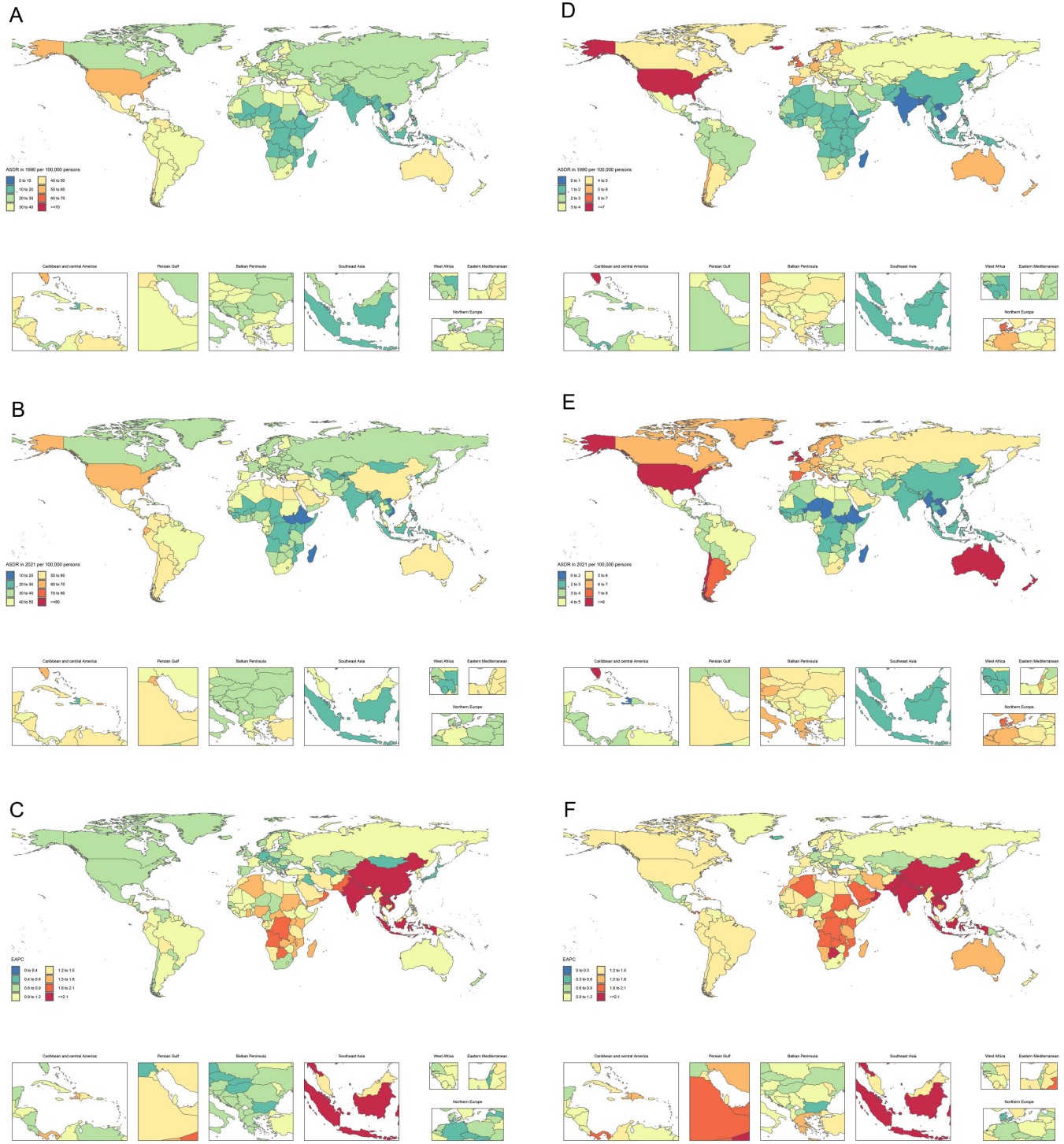

**Fig 3. The ASDR of early-onset OA attributable to high BMI per 100,000 population in 1990 and 2021, by country, along with ASDR trends from 1990 to 2021 as measured by the EAPC.** Note: A, The ASDR of early-onset knee OA attributable to high BMI in 1990, by country; B, The ASDR of early-onset knee OA attributable to high BMI in 2021, by country; C, The trend in ASDR of early-onset knee OA attributable to high BMI from 1990 to 2021; D, The ASDR of early-onset hip OA attributable to high BMI in 1990, by country; E, The ASDR of early-onset hip OA attributable to high BMI in 2021, by country; F, The trend in ASDR of early-onset hip OA attributable to high BMI from 1990 to 2021. Abbreviations: BMI, Body mass index; EAPC, estimated annual percentage change; ASDR, age-standardized disability-adjusted life years rate.

Early-onset knee osteoarthritis          Early-onset hip osteoarthritis

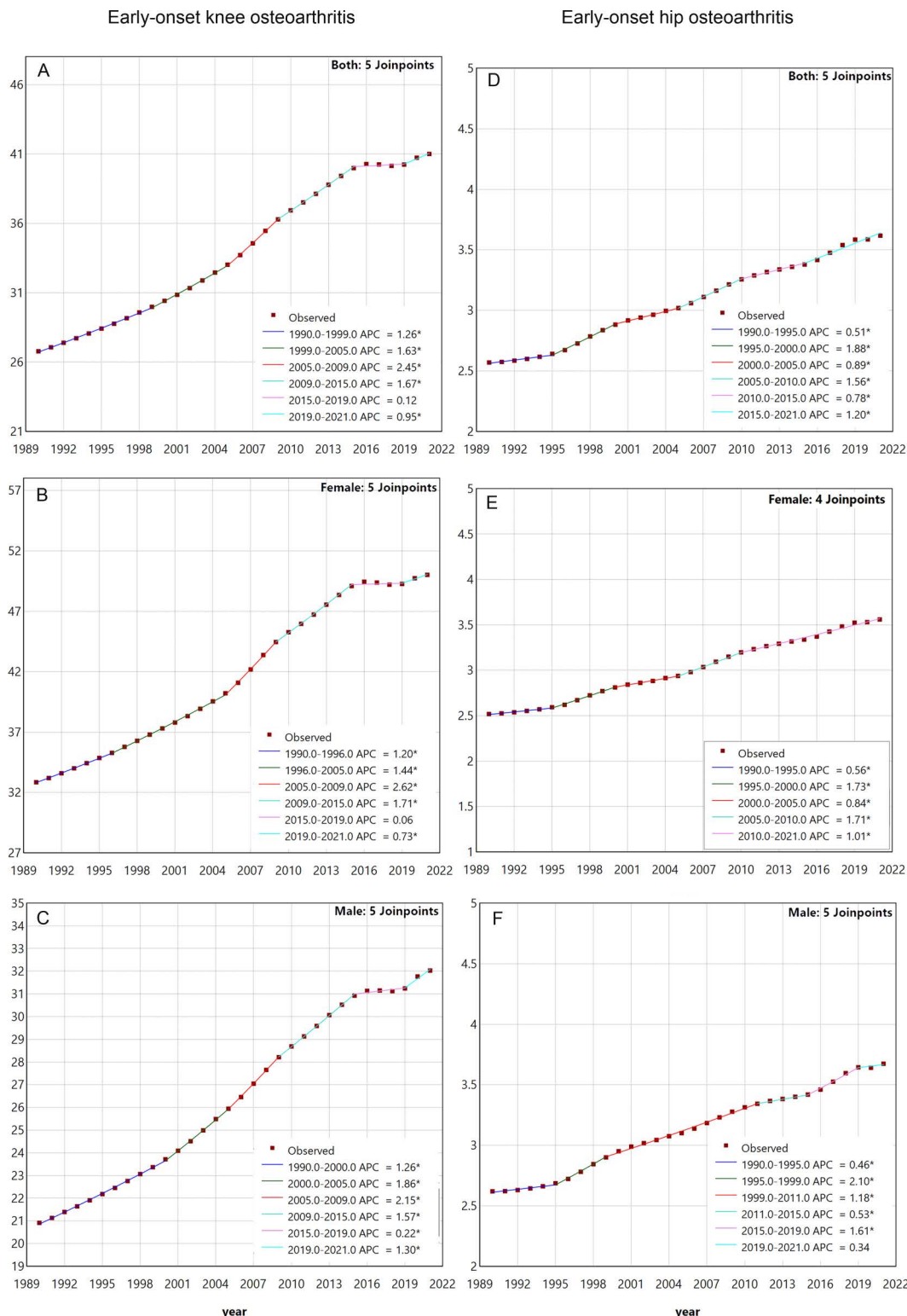

**Fig 4. Joinpoint regression analysis was conducted on the ASDR for early-onset OA attributable to high BMI.** Note: A, The joinpoint regression analysis for the ASDR of early-onset knee OA attributable to high BMI for all genders; B, The joinpoint regression analysis for the ASDR of early-onset knee OA attributable to high BMI for female; C, The joinpoint regression analysis for the ASDR of early-onset knee OA attributable to high BMI for male; D,

The joinpoint regression analysis for the ASDR of early-onset hip OA attributable to high BMI for all genders; E, The joinpoint regression analysis for the ASDR of early-onset hip OA attributable to high BMI for female; F, The joinpoint regression analysis for the ASDR of early-onset hip OA attributable to high BMI for male. Abbreviations: BMI, Body mass index; ASDR, age-standardized disability-adjusted life years rate.

variations were observed in the trends across different time periods. Specifically, the ASDR showed no significant change during the period from 2015 to 2019 (P=0.14), while it exhibited an increasing trend in all other time periods (P<0.05). In gender-stratified analyses, the trend for males showed an overall increase (AAPC=1.40%, 95% CI: 1.36%–1.43%), with increasing trends in all time periods (P<0.05). For females, the overall trend was also an increase (AAPC=1.37%, 95% CI: 1.34%–1.40%), but there were differences in the trends across time periods. The period from 2019 to 2021 showed no significant change (P=0.36), while other time periods exhibited increasing trends (P<0.05). For early-onset hip OA attributable to high BMI, an increasing trend was observed across all time periods (AAPC=1.14%, 95% CI: 1.02%–1.25%). In gender-stratified analyses, males demonstrated an overall increase (AAPC=1.10%, 95% CI: 0.99%–1.21%), with the exception of the period from 2019 to 2021, which showed no significant change (P=0.43); however, increasing trends were observed in other time periods (P<0.05). Females also demonstrated an overall increasing trend (AAPC=1.14%, 95% CI: 1.04%–1.23%), with increasing trends across all time periods (P<0.05). (S4 Table).

The results of the Age-Period-Cohort (APC) analysis are presented in Fig 5 and S5 Table. For early-onset knee OA attributable to high BMI, after controlling for period and birth cohort effects, the age effect had a significant impact on the DALYs. The DALYs risk consistently increased with age (S5 Table). After controlling for age and birth cohort effects, the period effect from 1992 to 2021 showed an increasing trend in DALYs risk. The highest ASDR was observed in the period 2019, with the rate ratio (RR) increasing by 1.44 times (S5 Table). After controlling for age and period effects, the RR for earlier birth cohorts was lower compared to later birth cohorts. The RR continued to increase from the 1938–1946 cohort to the 1983–1991 cohort (S5 Table). Gender-stratified analyses revealed that, at the same stage, the age effect had a greater impact on females than on males. The period and cohort effects on DALYs were similar in males and females (S5 Table). For early-onset hip OA caused by high BMI, after controlling for period and birth cohort effects, the age effect also had a significant impact on DALYs. The DALYs risk consistently increased with age (S5 Table). After controlling for age and birth cohort effects, the period effect from 1992 to 2021 showed a continuous increase in DALYs risk. The RR of DALYs in period 2019 increased by 1.39 times (S5 Table). After controlling for age and period effects, the RR for later birth cohorts was higher compared to earlier birth cohorts. The RR continued to rise from the 1938–1946 cohort to the 1983–1991 cohort (S5 Table). Gender-stratified analyses showed that, at the same stage, the age effect had a greater impact on males than on females. The period and cohort effects on females were stronger than those on males (S5 Table).

Compared to 1990, the DALYs increased in 2021. To quantify the impact of population aging, population growth, and epidemiological changes on the burden of disease over the past 32 years, we performed a decomposition analysis (Fig 6). Globally, for early-onset knee OA attributable to high BMI, population growth was the primary driver of the increase in DALYs. Specifically, aging, population growth, and epidemiological changes explained10.93%, 50.49%, and 38.58% of the increase in DALYs, respectively. In gender-stratified analyses, aging, population growth, and epidemiological changes explained 10.49%, 50.39%, and 39.15% of the increase in male DALYs, and 11.38%, 50.85%, and 37.76% of the increase in female global DALYs, respectively. Similarly, globally, population growth was also the main driver of the increase in DALYs for early-onset hip OA attributable to high BMI. Specifically, aging, population growth, and epidemiological changes explained 8.87%, 56.35%, and 34.78% of the increase in DALYs. In gender-stratified analyses, aging, population growth, and epidemiological changes explained 8.57%, 56.42%, and 35.01% of the increase in male DALYs, and 9.16%, 56.26%, and 34.58% of the increase in female DALYs, respectively (Fig 6, S6 Table).

From 1990 to 2021, we observed that the ASDR for early-onset knee OA attributable to high BMI increased with the SDI until an SDI value of approximately 0.7, after which it began to decline. The ASDR in Oceania, Central Latin America,

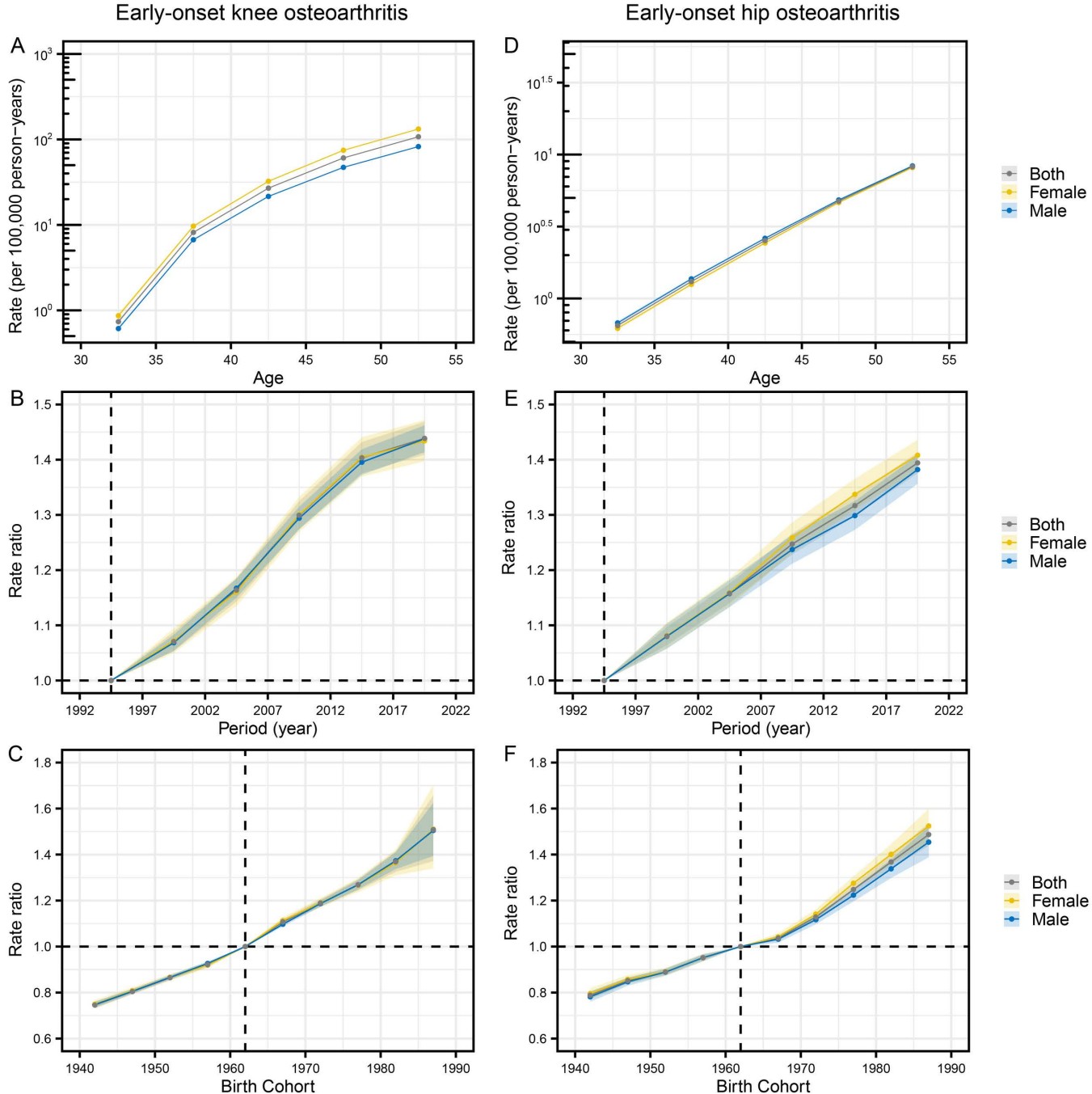

**Fig 5. The effects of age, period, and birth cohort on the relative risk of DALYs for early-onset OA attributable to high BMI.** Note: A-C, Early-onset knee OA attributable to high BMI; D-F, Early-onset hip OA attributable to high BMI. Abbreviations: BMI, Body mass index; DALYs, Disability-adjusted life years.

Andean Latin America, Southern Latin America, and High-Income North America was higher than expected based on their SDI during the period from 1990 to 2021 (Fig 7). At the national level, the ASDR for early-onset knee OA due to high BMI exhibited an M-shaped fluctuation with increasing SDI. Countries such as the Cook Islands, American Samoa, and Tonga

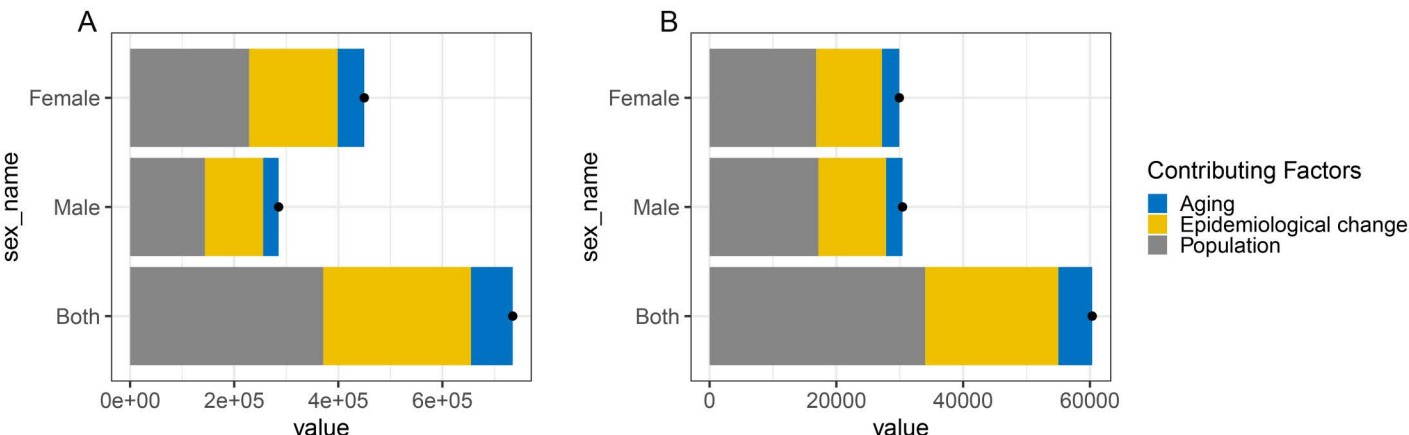

**Fig 6. Decomposition analysis of changes in the DALYs of early-onset OA attributable to high BMI, stratified by gender from 1990 to 2021.** Note: A, Early-onset knee OA attributable to high BMI; B, Early-onset hip OA attributable to high BMI. Abbreviations: BMI, Body mass index; DALYs, Disability-adjusted life years.

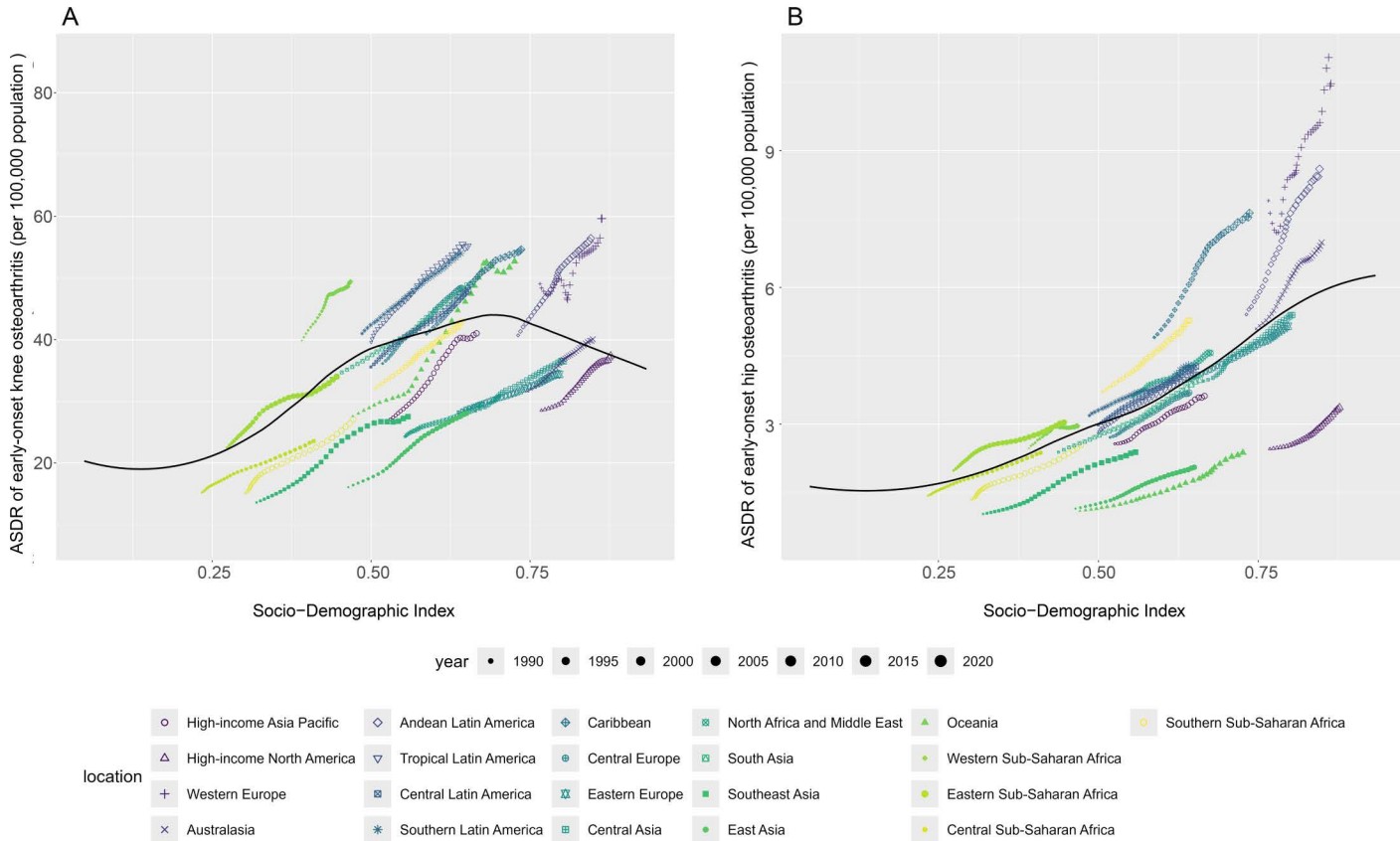

**Fig 7. The associations between the SDI and the ASDR of early-onset OA attributable to high BMI across 21 GBD regions.** Note: The ASDR of early-onset OA attributable to high BMI across 21 GBD regions from 1990 to 2021, categorized by the SDI. Each region is represented by 32 points, illustrating the observed ASDR trends over time. The solid line represents the expected ASDR values based on the SDI and overall disease incidence across all regions. Regions positioned above the solid line indicate a higher-than-expected burden, while those below the line indicate a lower-than-expected burden. A, early-onset knee OA attributable to high BMI; B, early-onset hip OA attributable to high BMI. Abbreviations: BMI, Body mass index; ASDR, age-standardized disability-adjusted life years rate; GBD, Global Burden of Disease; SDI, Sociodemographic Index.

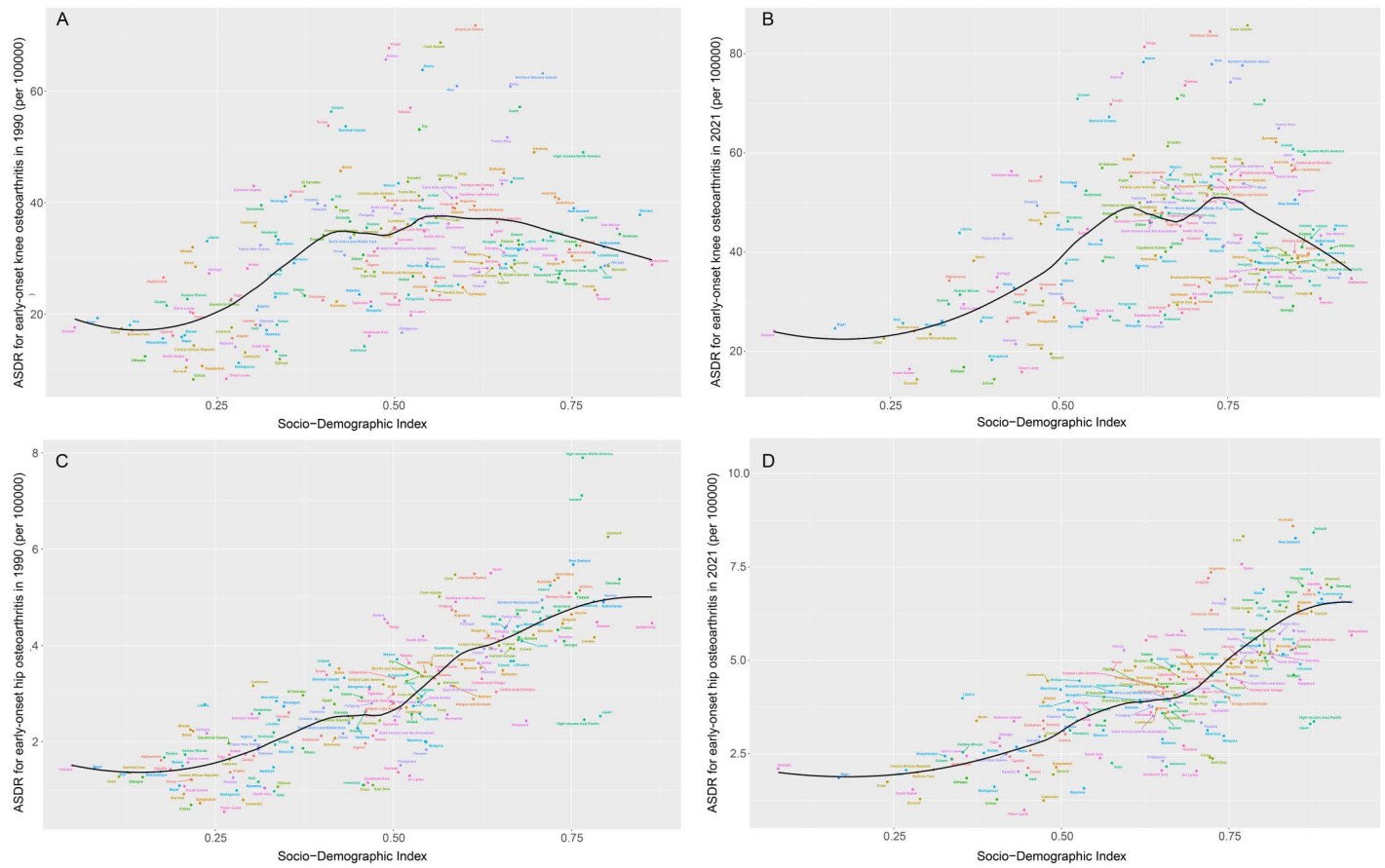

**Fig 8. The associations between the SDI and the ASDR of early-onset OA attributable to high BMI across 204 countries.** Note: A, The association between the SDI and the ASDR of early-onset knee OA attributable to high BMI across 204 countries in 1990; B, The association between the SDI and the ASDR of early-onset knee OA attributable to high BMI across 204 countries in 2021; C, The association between the SDI and the ASDR of early-onset hip OA attributable to high BMI across 204 countries in 1990; D, The association between the SDI and the ASDR of early-onset hip OA attributable to high BMI across 204 countries in 2021. Abbreviations: BMI, Body mass index; ASDR, age-standardized disability-adjusted life years rate; GBD, Global Burden of Disease; SDI, Sociodemographic Index.

had a disease burden far exceeding expectations (Fig 8). Gender-stratified analyses showed similar trends in both males and females (S3–S6 Figs). Through cross-national inequality analysis, we found significant absolute and relative inequalities in the disease burden of early-onset knee OA attributable to high BMI associated with SDI. Countries with higher levels of social development shouldered a higher disease burden. The Slope Index of Inequality (SII) indicated that the difference in ASDR between the highest and lowest SDI countries in 1990 was 42.27, and this gap increased to 86.42 by 2021. Furthermore, the concentration index (CI), as a measure of relative gradient inequality, decreased from 0.29 in 1990 to 0.24 in 2021. These results suggest that, although absolute health inequality in the disease burden of early-onset knee OA caused by obesity increased from 1990 to 2021, relative inequality decreased.

The ASDR for early-onset hip OA caused by high BMI increased with the SDI. The ASDR in Western Sub-Saharan Africa, Oceania, Tropical Latin America, Central Latin America, Southern Latin America, the Caribbean, Andean Latin America, Southern Sub-Saharan Africa, High-Income North America, Western Europe, and Australasia was higher than expected based on their SDI during the period from 1990 to 2021 (Fig 7). At the national level, the ASDR for early-onset hip OA due to high BMI also increased with rising SDI. Countries and regions such as Australia, Iceland, and New Zealand

had a disease burden far exceeding expectations (Fig 8). Gender-stratified analyses showed similar trends in both males and females (S3–S6 Figs). Through cross-national inequality analysis, we found significant absolute and relative inequalities in the disease burden of early-onset hip OA caused by high BMI associated with SDI. Countries with higher levels of social development bore a higher disease burden. The SII showed that the difference in ASDR between the highest and lowest SDI countries in 1990 was 5.42, and this gap increased to 12.00 by 2021. Additionally, the concentration index, as a measure of relative gradient inequality, decreased from 0.39 in 1990 to 0.31 in 2021. These results suggest that, although absolute health inequality in the disease burden of early-onset hip OA caused by high BMI increased from 1990 to 2021, relative inequality decreased.

Fig 9 and S7 Table show that the global burden of early-onset OA caused by high BMI is projected to increase steadily over the next 15 years. By 2036, the ASDR for early-onset knee OA due to high BMI is expected to reach 46.51 cases per 100,000 population, with the disease burden in female continuing to exceed that in male. Specifically, the ASDR for female in 2036 is projected to be 48.63 cases per 100,000 population, compared to 36.94 cases per 100,000 population for male. The disease burden of early-onset hip OA influenced by high BMI also shows a steady upward trend. By 2036, the ASDR for early-onset hip OA is expected to reach 4.15 cases per 100,000 population, with male expected to bear a higher burden than female. Specifically, the ASDR for female in 2036 is projected to be 4.09 cases per 100,000 population, while for male, it is expected to be 4.22 cases per 100,000 population.

## 4 Discussion

This study aims to explore the global burden of early-onset OA caused by high BMI from 1990 to 2021.The traditional view holds that countries with higher SDI levels and higher income typically experience lower disease burdens due to more advanced healthcare systems and more effective health policy execution. However, the results of this study reveal an opposite trend: the burden of early-onset OA due to high BMI is positively correlated with SDI levels. In high SDI regions, the burden of early-onset OA caused by high BMI is more severe, which may be linked to the higher prevalence of obesity in high-income countries. A study covering 70 low, middle, and high-income countries indicated that the average BMI of populations in low-income countries is below 25.0, while most high-income countries have an average BMI above 25.0 [33]. In recent years, obesity rates among children and adolescents in most high-income countries have stabilized but remain at high levels, while the number of obese children and adolescents in low- and middle-income countries is increasing [34]. Persistent obesity during childhood and adolescence can lead to long-term joint overload, increasing the risk of early cartilage degeneration. Additionally, sports injuries make obese individuals in this age group more prone to developing OA in their youth or middle age [35]. The multifactorial mechanisms of obesity further exacerbate the burden of early-onset OA. Obese individuals experience a significant increase in the levels of inflammatory mediators, such as interleukin-6 and tumor necrosis factor-alpha, secreted by adipose tissue. These mediators not only promote systemic inflammation but also directly affect joint tissues, leading to the onset and progression of OA [36]. Moreover, being overweight or obese significantly increases the mechanical load on joints and alters their biomechanical properties [37]. In the case of the knee joint, the weight of an obese individual causes a shift in body mass distribution, which can trigger varus or valgus deformities of the knee. Under these deformities, the way the joint bears load changes dramatically, concentrating pressure on specific areas, which greatly increases the risk of damage to cartilage and bone tissues [38]. This is particularly true for obese youth and middle-aged individuals, as the pressure on their joints is further amplified during daily physical labor or exercise. This sustained high-pressure state easily leads to cartilage degeneration, damage, and the destruction of normal joint structure and function, ultimately resulting in OA. Additionally, even in developed countries such as the United States and the United Kingdom, current treatments for early-onset OA and strategies to delay disease progression remain inadequate, leading to a continued increase in the number of early-onset OA cases caused by obesity. The differences between countries and regions with varying SDI levels—including public awareness of early-onset OA, prevention awareness of risk factors, healthcare system utilization, and the capacity for precise diagnosis and

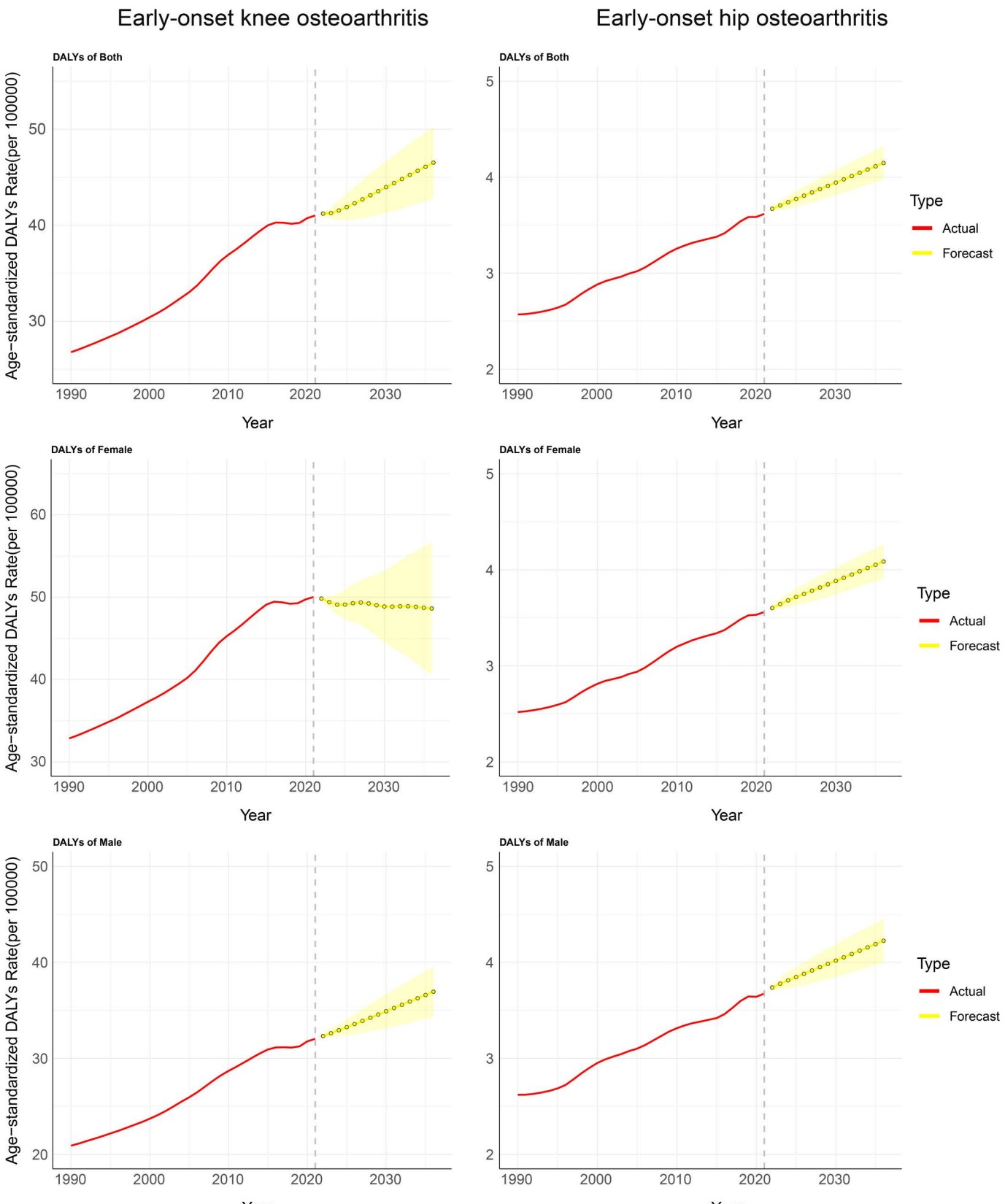

Early-onset knee osteoarthritis — Early-onset hip osteoarthritis

**Fig 9. The projected ASDR of early-onset OA attributable to high BMI over the next 15 years.** Note: A, The projected ASDR of early-onset knee OA attributable to high BMI for all genders; B, The projected ASDR of early-onset knee OA attributable to high BMI for female; C, The projected ASDR of early-onset knee OA attributable to high BMI for male; D, The projected ASDR of early-onset hip OA attributable to high BMI for all genders. E, The projected ASDR of early-onset hip OA attributable to high BMI for female; F, The projected ASDR of early-onset hip OA attributable to high BMI for male. Abbreviations: BMI, Body mass index; ASDR, age-standardized disability-adjusted life years rate.

treatment—may further exacerbate inequalities in disease burden. These disparities indicate that targeted interventions tailored to countries with different SDI levels are necessary to effectively address the burden of early-onset OA caused by high BMI.

Between 1990 and 2021, the disease burden of early-onset OA due to high BMI has gradually increased. Previous studies have also indicated a continuous rise in the disease burden of early-onset OA [6]. Other studies examining OA disease burden have shown an increase across all age groups from 1990 to 2021 [20,39,40]. Through joinpoint regression analysis, this study further reveals the phase changes within the overall trend. Notably, the ASDR for early-onset knee OA caused by high BMI showed a period of stability (such as 2015–2019). This phenomenon may be closely related to significant lifestyle improvements and advances in medical technology during this period. Advanced diagnostic techniques, such as high-resolution imaging devices, have enabled earlier and more accurate detection of subtle joint lesions, facilitating early diagnosis and prevention. Additionally, socioeconomic development has promoted the widespread dissemination of public health knowledge. People are now more easily able to access information on weight loss, optimizing exercise routines, and avoiding joint damage, leading to proactive adoption of healthy lifestyles. These factors may have collectively enhanced public awareness of early-onset OA prevention, partially slowing the growth of the disease burden. However, the impact of high BMI on early-onset OA extends beyond physical health and involves multidimensional issues, including economic and social concerns. High BMI and related diseases not only cause significant physical pain and severe limitations in mobility but also indirectly lead to a series of economic burdens. Patients often face high medical costs, and the reduced ability to work, or even loss of productivity, leads to a significant economic loss. As the condition worsens, many patients may require specialized care, further increasing care costs. According to relevant studies, disease-related expenditures due to obesity account for between 0.7% and 17.8% of total healthcare spending, and between 0.05% and 2.42% of GDP [41]. Therefore, given the dynamic changes in the disease burden over different periods, there is an urgent need to adjust healthcare prevention and management policies in a timely manner to better adapt to these changes and more effectively address this increasingly severe public health challenge.

The Age-Period-Cohort (APC) model is a widely used statistical tool for analyzing epidemiological data on diseases, allowing for the separation of the effects of age, period, and birth cohort on disease patterns [42]. In this study, the APC model revealed that the DALYs risk of early-onset OA due to high BMI significantly increases with age. The age effect suggests that aging is one of the primary risk factors for the increasing burden of OA. As people age, particularly women, joint cartilage gradually undergoes degenerative changes due to decreased estrogen levels. The activity of chondrocytes decreases, and the ability to synthesize matrix diminishes, leading to thinning of cartilage [43,44]. Additionally, the reduction in muscle mass and function in elderly patients further weakens the protective effect on the joints, subjecting them to higher biomechanical stress, making them more prone to damage [45,46]. The period effect analysis shows that the relative risk of early-onset OA due to high BMI has increased annually over time. This is likely linked to changes in social structure and lifestyle. From a gender perspective, the period effect had a significantly greater impact on men than on women, particularly in the case of early-onset hip OA due to high BMI. With urbanization and industrialization, more men engage in physically demanding occupations, such as construction, which significantly increases the load on the hip joint, making the relative risk of hip OA higher in men than in women [47]. The birth cohort effect reveals that later-born cohorts face a higher risk of early-onset OA due to high BMI. This trend is closely related to changes

in lifestyle during social development. People born later live in environments with richer material conditions, where abundant and diverse food supplies have significantly increased obesity rates. At the same time, while younger generations have greater health awareness and physical activity is widely promoted, joint injuries from exercise may later develop into OA [48]. The risk trends revealed by the APC model indicate that, while improvements in modern lifestyle have brought numerous health benefits, they have also introduced new health challenges. Aging and lifestyle changes have played a significant role in exacerbating the burden of OA. Therefore, how to reduce the risk of OA due to lifestyle changes while benefiting from improved material conditions and increased health awareness is an important area for future focus and in-depth research.

The ARIMA model predicts that the disease burden of early-onset OA due to high BMI will continue to increase over the next 15 years. Population growth and changes in economic structure are expected to remain the primary drivers of this rising disease burden. The high prevalence of obesity among younger populations not only significantly increases the risk of early-onset OA but also extends the course of the disease, resulting in a higher medical burden. As the number of patients increases, the demand for medical resources will surge significantly. For patients with advanced OA, joint replacement surgery is a common treatment, but it is costly, and the lifespan of artificial joints is limited. For instance, for every 5-year decrease in the age at which knee replacement is performed, the risk of revision surgery increases by 10% [49]. This trend not only intensifies the financial burden on patients but also poses significant challenges for the allocation of medical resources. The forecast further indicates that the disease burden of early-onset knee OA due to high BMI will continue to be higher in women than in men, while the disease burden of early-onset hip OA due to high BMI will remain higher in men than in women. This gender disparity may be closely related to future changes in social structure. Over the next 15 years, as urbanization continues, the proportion of men in physically demanding industries may still be high, leading to greater hip joint load and injury risk. To address the rising disease burden and gender disparities in the future, more precise intervention measures need to be implemented. For women: focus on weight control and hormone management, and use early interventions to delay degenerative changes in the joints. For men: enhance occupational protection measures to reduce joint damage caused by high-intensity labor, while promoting scientific exercise knowledge to reduce the incidence of exercise-related injuries.

This study systematically analyzed the global disease burden of early-onset OA attributable to high BMI from 1990 to 2021, revealing an upward trend and disparities among countries and regions. These findings provide comprehensive and high-quality data support for global health research and public health policymaking. However, we also recognize several limitations in this study. Firstly, this study found that the confidence intervals for DALYs in some countries included negative values, even though the actual DALYs values were non-negative. This reflects the estimation bias of statistical models when data are insufficient. The analysis is based on the GBD 2021 database, which integrates diverse data sources using Bayesian statistical methods. While it serves as a large-scale meta-analysis, the heterogeneity of data sources and variations in correction methods may introduce additional uncertainties, potentially affecting the precision of the results. Moreover, the database's estimates of disease burden may not fully reflect the specific circumstances of certain countries or regions. For instance, in China, the unique geographic distribution and population characteristics might result in ecological fallacies, limiting the direct applicability of the findings. Future research should aim to overcome these limitations and explore the complexities of the relationship between high BMI and early-onset OA. This could be achieved through the following approaches: By refining country- and region-specific analyses, future studies can enhance the accuracy of model estimates and better reflect local realities. Long-term tracking of individuals' lifestyles and genetic information, combined with comprehensive disease surveillance data, can help uncover causal links between high BMI and early-onset OA. Tailoring preventive strategies based on the specific circumstances of different countries and regions, using disease monitoring and detailed data analysis, can enhance intervention effectiveness. By addressing the current limitations and expanding the scope of influencing factors, future research can significantly advance the field of orthopedics, particularly in understanding high BMI-related early-onset OA. This will not only

provide a solid theoretical foundation for disease prevention but also contribute to the optimization of global public health policies.

## Supporting information

**S1 Appendix.  Overview of the data screening of early-onset osteoarthritis attributable to high BMI in the 2021 Global Burden of Disease study.**
(DOCX)

**S1 Fig.  The ASDR of early-onset OA attributable to high BMI among males per 100,000 population in 1990 and 2021, by country, along with ASDR trends from 1990 to 2021 as measured by the EAPC.** Note: A, The ASDR of early-onset knee osteoarthritis attributable to high BMI among males in 1990, by country; B, The ASDR of early-onset knee osteoarthritis attributable to high among males BMI in 2021, by country; C, The trend in ASDR of early-onset knee osteoarthritis attributable to high BMI among males from 1990 to 2021; D, The ASDR of early-onset hip osteoarthritis attributable to high BMI among males in 1990, by country; E, The ASDR of early-onset hip osteoarthritis attributable to high BMI among males in 2021, by country; F, The trend in ASDR of early-onset hip osteoarthritis attributable to high BMI among males from 1990 to 2021. Abbreviations: BMI, Body mass index; EAPC, estimated annual percentage change; ASDR, age-standardized disability-adjusted life years rate.
(DOCX)

**S2 Fig.  The ASDR of early-onset OA attributable to high BMI among female per 100,000 population in 1990 and 2021, by country, along with ASDR trends from 1990 to 2021 as measured by the EAPC.** Note: A, The ASDR of early-onset knee osteoarthritis attributable to high BMI among female in 1990, by country; B, The ASDR of early-onset knee osteoarthritis attributable to high among female BMI in 2021, by country; C, The trend in ASDR of early-onset knee osteoarthritis attributable to high BMI among female from 1990 to 2021; D, The ASDR of early-onset hip osteoarthritis attributable to high BMI among female in 1990, by country; E, The ASDR of early-onset hip osteoarthritis attributable to high BMI among female in 2021, by country; F, The trend in ASDR of early-onset hip osteoarthritis attributable to high BMI among female from 1990 to 2021. Abbreviations: BMI, Body mass index; EAPC, estimated annual percentage change; ASDR, age-standardized disability-adjusted life years rate.
(DOCX)

**S3 Fig.  The association between the ASDR of male early-onset OA attributable to high BMI among male and the SDI across the 21 GBD regions.** Note: A, Early-onset knee osteoarthritis attributable to high BMI; B, Early-onset hip osteoarthritis attributable to high BMI. Abbreviations: BMI, Body mass index; ASDR, age-standardized disability-adjusted life years rate; GBD, Global Burden of Disease; SDI, Sociodemographic Index.
(DOCX)

**S4 Fig.  The association between the ASDR of female early-onset OA attributable to high BMI among female and the SDI across the 21 GBD regions.** Note: A, Early-onset knee osteoarthritis attributable to high BMI; B, Early-onset hip osteoarthritis attributable to high BMI. Abbreviations: BMI, Body mass index; ASDR, age-standardized disability-adjusted life years rate; GBD, Global Burden of Disease; SDI, Sociodemographic Index.
(DOCX)

**S5 Fig.  The association between the SDI and the ASDR of male early-onset OA attributable to high BMI among male across 204 countries.** Note: A, The association between the SDI and the ASDR of early-onset knee osteoarthritis attributable to high BMI among male across 204 countries in 1990; B, The association between the SDI and the ASDR of early-onset knee osteoarthritis attributable to high BMI among male across 204 countries in 2021; C, The association

between the SDI and the ASDR of early-onset hip osteoarthritis attributable to high BMI among male across 204 countries in 1990; D, The association between the SDI and the ASDR of early-onset hip osteoarthritis attributable to high BMI among male across 204 countries in 2021. Abbreviations: BMI, Body mass index; ASDR, age-standardized disability-adjusted life years rate; SDI, Sociodemographic Index.
(DOCX)

**S6 Fig. The association between the SDI and the ASDR of female early-onset OA attributable to high BMI among female across 204 countries.** Note: A, The association between the SDI and the ASDR of early-onset knee osteoarthritis attributable to high BMI among female across 204 countries in 1990; B, The association between the SDI and the ASDR of early-onset knee osteoarthritis attributable to high BMI among female across 204 countries in 2021; C, The association between the SDI and the ASDR of early-onset hip osteoarthritis attributable to high BMI among female across 204 countries in 1990; D, The association between the SDI and the ASDR of early-onset hip osteoarthritis attributable to high BMI among female across 204 countries in 2021. Abbreviations: BMI, Body mass index; ASDR, age-standardized disability-adjusted life years rate; SDI, Sociodemographic Index.
(DOCX)

**S1 Table. The ASDR of early-onset OA attributed to high BMI in 1990 and 2021 for male by SDI quintiles and by GBD regions, with EAPC from 1990 to 2021.**
(DOCX)

**S2 Table. The ASDR of early-onset OA attributed to high BMI in 1990 and 2021 for female by SDI regions and by GBD regions, with EAPC from 1990 to 2021.**
(DOCX)

**S3 Table. The ASDR of early-onset OA attributed to high BMI in 1990 and 2021 for both sexes by countries, with EAPC from 1990 to 2021.**
(DOCX)

**S4 Table. Joinpoint regression analysis was conducted on the ASDR for both sexes.**
(DOCX)

**S5 Table. The effects of age, period, and birth cohort on the relative risk of DALYs for early-onset OA attributable to high BMI.**
(DOCX)

**S6 Table. Decomposition analysis of changes in the DALYs of early-onset OA attributable to high BMI, stratified by gender from 1990 to 2021.**
(DOCX)

**S7 Table. The projected ASDR of early-onset OA attributable to high BMI over the next 15 years.**
(DOCX)

## Acknowledgments

Heartfelt thanks are extended to the Institute for Health Metrics and Evaluation (IHME) at the University of Washington. We also extend our high esteem and heartfelt thanks to all the collaborators of the GBD (Global Burden of Disease Study). It is important to note that the views and opinions expressed in this article are those of the authors and do not represent the official position or views of the organizations to which they belong.

## Author contributions

**Data curation:** Binbin Zhang, Qi Yan.

**Formal analysis:** Bin Dou, Chuan Lu, Dawa Zhaxi, Yu Wang, Jiale Xu.

**Funding acquisition:** Wenzuo Gu.

**Methodology:** Binbin Zhang, Chuan Lu, Qi Yan, Kewen Li.

**Software:** Binbin Zhang, Chuan Lu, Qi Yan, Dawa Zhaxi, Yu Wang, Jiale Xu, Kewen Li.

**Supervision:** Kewen Li.

**Writing – original draft:** Binbin Zhang, Bin Dou, Kewen Li.

**Writing – review & editing:** Binbin Zhang, Bin Dou, Wenzuo Gu, Kewen Li.

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
