## [Decision Letter · Decision Letter 0]

PONE-D-25-07975The global, regional, and national burden of early-onset osteoarthritis attributable to high BMI from 1990 to 2021, along with projections for 2036: A systematic analysis of the 2021 Global Burden of Disease Study.PLOS ONE

Dear Dr. Li,

Thank you for submitting your manuscript to PLOS ONE. After careful consideration, we feel that it has merit but does not fully meet PLOS ONE’s publication criteria as it currently stands. Therefore, we invite you to submit a revised version of the manuscript that addresses the points raised during the review process.

Please make peer-to-peer modifications to the reviewer's comments.

We look forward to receiving your revised manuscript.

Kind regards,

Qian Wu

Academic Editor

PLOS ONE

Journal Requirements:

[This study was funded by 2024 Provincial and ministerial joint construction of Central Asia high disease causes and prevention of the national key laboratory-Qinghai Workstation Joint Fund�SKL-HIDCA-2024-QH1�The funders had no role in study design, data collection and analysis, decision to publish, or preparation of the manuscript.].

3. Please upload a new copy of Figures 1, 2, 4, 5, 7, 8, and 9 as the details are not clear. Please follow the link for more information: https://blogs.plos.org/plos/2019/06/looking-good-tips-for-creating-your-plos-figures-graphics/
https://blogs.plos.org/plos/2019/06/looking-good-tips-for-creating-your-plos-figures-graphics/

4. We note that Figure 3, and Supplementary Figures 1 & 2 in your submission contain [map/satellite] images which may be copyrighted. All PLOS content is published under the Creative Commons Attribution License (CC BY 4.0), which means that the manuscript, images, and Supporting Information files will be freely available online, and any third party is permitted to access, download, copy, distribute, and use these materials in any way, even commercially, with proper attribution. For these reasons, we cannot publish previously copyrighted maps or satellite images created using proprietary data, such as Google software (Google Maps, Street View, and Earth). For more information, see our copyright guidelines: http://journals.plos.org/plosone/s/licenses-and-copyright.

              1. You may seek permission from the original copyright holder of Figure 3, and Supplementary Figures 1 & 2  to publish the content specifically under the CC BY 4.0 license. 

Reviewers' comments:

Reviewer's Responses to Questions

**Comments to the Author**

1. Is the manuscript technically sound, and do the data support the conclusions?

Reviewer #1: Yes

Reviewer #2: Yes

Reviewer #3: Yes

2. Has the statistical analysis been performed appropriately and rigorously? 

Reviewer #1: Yes

Reviewer #2: Yes

Reviewer #3: Yes

3. Have the authors made all data underlying the findings in their manuscript fully available?

Reviewer #1: Yes

Reviewer #2: Yes

Reviewer #3: Yes

4. Is the manuscript presented in an intelligible fashion and written in standard English?

Reviewer #1: Yes

Reviewer #2: Yes

Reviewer #3: Yes

5. Review Comments to the Author

Reviewer #1: The article is well written. Few suggestions are there for consideration.

1. please try to frame title in PI(E)COS format.

2. SDI quintile classification is The 204 countries and territories covered in the Global Burden of Disease (GBD) 2019 were divided into five SDI quintiles: low SDI (0 to 0.4547), low-middle SDI (0.4547 to 0.6076), middle SDI (0.6076 to 0.7905), high-middle SDI (0.7905 to 0.8051), and high SDI (0.8051 to 1). It is a bit different from that mentioned here. Please mention the reference.

3. In ARIMA, is here d have been used for testing difference between different groups?

4. In discussion section repetition of current study results are not needed to mention.

5. References to write down in uniform manner. Page no. to be like, say in ref. no. 4 531-41.

Reviewer #2: 1. It is recommended that the abstract be slightly condensed and simplified to enhance readability.

2. The graphs and tables are well-designed; however, for certain countries with negative confidence intervals, the authors may consider commenting on the statistical accuracy of those estimates.

3. The reference section could be further strengthened by including more recent sources from 2022–2024, although the current references are valid and relevant.

Reviewer #3: Thank You for submitting your work. The methodology is rigourous and the topic is interesting. The forecast modelling adds an interesting aspect to the study. Excellent work on the manuscript overall.

6. PLOS authors have the option to publish the peer review history of their article (what does this mean? ). If published, this will include your full peer review and any attached files.

**Do you want your identity to be public for this peer review?** For information about this choice, including consent withdrawal, please see our Privacy Policy .

Reviewer #1: **Yes: ** Dr Satabdi Mitra

Reviewer #2: No

Reviewer #3: No

---

## [Author Response · Author response to Decision Letter 1]

29 May 2025

Dear editor and reviewers:

Thank you very much for your e-mail dated on May 28, 2025, informing us the editorial decision on our manuscript (PONE-D-25-07975). We would like to express our gratitude to you and the editor, as well as the anonymous reviewers for the time and effort spent in processing and improving our paper. The constructive comments from you and the reviewers are very helpful for the improvement of our paper. This is to confirm that the paper has been duly revised in accordance with the comments made by you and the reviewers, which we would like to resubmit as a possible publication in the PLOS ONE . Attached files are the detailed responses to you and the reviewers, and the duly revised manuscript. Once again, we sincerely thank you for the time and effort you have spent and are going to spend in processing our paper. We look forward to hearing from you regarding its disposition.

We have carefully addressed each of the editorial concerns and have made the necessary revisions as outlined below. For each comment, we provide a detailed response and describe the specific revisions made to the manuscript.

Responses to Editor

Manuscript Code: PONE-D-25-07975

The global, regional, and national burden of early-onset osteoarthritis attributable to high BMI from 1990 to 2021, along with projections for 2036: A systematic analysis of the 2021 Global Burden of Disease Study

1.Comment: Please ensure that your manuscript meets PLOS ONE's style requirements, including those for file naming.

Response: Thank you for your valuable comment. We have now followed the Plos One template according to the submission guidelines and PLOS ONE's style requirements, including those for file naming.

2.Comment: Thank you for stating in your Funding Statement: [This study was funded by 2024 Provincial and ministerial joint construction of Central Asia high disease causes and prevention of the national key laboratory-Qinghai Workstation Joint Fund�SKL-HIDCA-2024-QH1�The funders had no role in study design, data collection and analysis, decision to publish, or preparation of the manuscript.]. Please provide an amended statement that declares *all* the funding or sources of support (whether external or internal to your organization) received during this study, as detailed online in our guide for authors at http://journals.plos.org/plosone/s/submit-now. Please also include the statement “There was no additional external funding received for this study.” in your updated Funding Statement. Please include your amended Funding Statement within your cover letter. We will change the online submission form on your behalf.

Response: This work was supported by the 2024 Provincial-Ministerial Jointly-Built National Key Laboratory of Etiology and Prevention of High-Incidence Diseases in Central Asia-Qinghai Workstation Joint Fund (SKL-HIDCA-2024-QH1) for research design, data collection and analysis, and decision to publish or write manuscripts. There was no additional external funding received for this study.

3.Comment: Please upload a new copy of Figures 1, 2, 4, 5, 7, 8, and 9 as the details are not clear.

Response: Thank you for your valuable comment. We have identified unclear details in Figures 1, 2, 4, 5, 7, 8, and 9. We have redrawn all figures using the original data. At the same time, we have verified the accuracy of these figures. The revised figures have been re-uploaded.

4.Comment: We note that Figure 3, and Supplementary Figures 1 & 2 in your submission contain [map/satellite] images which may be copyrighted. All PLOS content is published under the Creative Commons Attribution License (CC BY 4.0), which means that the manuscript, images, and Supporting Information files will be freely available online, and any third party is permitted to access, download, copy, distribute, and use these materials in any way, even commercially, with proper attribution. For these reasons, we cannot publish previously copyrighted maps or satellite images created using proprietary data, such as Google software (Google Maps, Street View, and Earth).

Response: Thank you for your questions regarding the map images in Fig 3 and and Supplementary Figures 1 & 2. Please find our responses below:

a)Source of Map Images:

The map images in Fig 3 and and Supplementary Figures 1 & 2 were created using the ggmap and maps R package. Based on ggplot2, it effectively integrates map data and spatial visualization. With it, you can access online maps such as OpenStreetMap. This data is open-source and available at https://cran.r-project.org/web/packages/ggmap/ggmap.pdf.

b) Copyright Status:

To our knowledge, the map data from the ggmap and maps R package is in the public domain and therefore not subject to copyright restrictions. The map data is widely used for world mapping and is explicitly designed for open, unrestricted use. We believe no additional copyright permission is required.

Please let us know if any further clarification is required, or if additional details or documentation are needed to satisfy PLOS ONE’s requirements.

Response: Thank you for your valuable comments. We have updated seven new references. Based on the reviewers' comments, we realized the necessity of citing references from 2022 to 2024 as much as possible. Therefore, by reviewing the latest literature, we updated references 3, 4, 10, 11, 21, 42, and 43.

Responses to Reviewer #1:

Manuscript Code: PONE-D-25-07975

The global, regional, and national burden of early-onset osteoarthritis attributable to high BMI from 1990 to 2021, along with projections for 2036: A systematic analysis of the 2021 Global Burden of Disease Study

1. Comment: Please try to frame title in PI(E)COS format.

Response: Thank you for your valuable suggestion regarding the title structure. We fully appreciate the importance of framing the title in the PI(E)COS format to enhance clarity and alignment with academic reporting standards. In response to your feedback, we have revised the title to explicitly incorporate the key elements of Population, Exposure, Outcome, Time, and Study Design.

Revised Title:

Global, regional, and national burden of early-onset osteoarthritis attributable to high BMI: 1990–2021 estimates and 2036 projections from the global burden of disease study.

2.Comment: SDI quintile classification is The 204 countries and territories covered in the Global Burden of Disease (GBD) 2019 were divided into five SDI quintiles: low SDI (0 to 0.4547), low-middle SDI (0.4547 to 0.6076), middle SDI (0.6076 to 0.7905), high-middle SDI (0.7905 to 0.8051), and high SDI (0.8051 to 1). It is a bit different from that mentioned here. Please mention the reference.

Response: Thank you for your valuable suggestions. For GBD 2021, the scope of SDI calculations has been expanded to include 1,075 countries and subnational regions, covering the period from 1950 to 2021. The latest data is available at https://ghdx.healthdata.org/record/global-burden-disease-study-2021-gbd-2021-socio-demographic-index-sdi-1950%E2%80%932021. For this reason, we have updated Reference 21 to: Global Burden of Disease Collaborative Network. Global Burden of Disease Study 2021 (GBD 2021) Socio-Demographic Index (SDI) 1950–2021. Seattle, United States of America: Institute for Health Metrics and Evaluation (IHME), 2024.

3.Comment: In ARIMA, is here d have been used for testing difference between different groups?

Response: Thank you for your thoughtful question regarding the use of the parameter d in the ARIMA model. We appreciate the opportunity to clarify this point. In the ARIMA framework, the parameter d (integrated order) specifically refers to the number of differences applied to the time series data to achieve stationarity, which is a fundamental assumption for time series modeling. This is distinct from testing differences between groups, which typically involves statistical methods such as t-tests, ANOVA, or regression models with categorical covariates. In our study, the ARIMA model was used to: Forecast trends in the burden of early-onset osteoarthritis attributable to high BMI over time (1990–2021 and projections to 2036). The parameter d was selected based on stationarity tests of the time series data (e.g., ADF test p-value < 0.05 for the differenced series), ensuring the model’s validity for temporal trend analysis.

4.Comment: In discussion section repetition of current study results are not needed to mention.

Response: We appreciate the reviewer’s suggestion to streamline the discussion section. In the revised manuscript, we have removed direct repetition of study results from the discussion section (e.g., specific ASDR values and trend percentages) and instead focused on interpreting these findings in the context of underlying mechanisms, comparative literature, and public health implications.

5.Comment: References to write down in uniform manner. Page no. to be like, say in ref. no. 4 531-41.

Response: We thank the reviewer for highlighting the need for uniform reference formatting. All references have been standardized to follow the journal’s citation guidelines, with page numbers presented in full and consistent use of Vancouver style for journal names, volumes, and issue numbers.

Responses to Reviewer #2

Manuscript Code: PONE-D-25-07975

The global, regional, and national burden of early-onset osteoarthritis attributable to high BMI from 1990 to 2021, along with projections for 2036: A systematic analysis of the 2021 Global Burden of Disease Study

1.Comment: It is recommended that the abstract be slightly condensed and simplified to enhance readability.

Response: We appreciate the reviewer’s suggestion to enhance the abstract’s readability. The abstract has been condensed by removing redundant background details and streamlining methodology descriptions. Key revisions include: 1) The Methods section has been streamlined to state: Data from the Global Burden of Disease 2021 (GBD) study were analyzed to assess early-onset osteoarthritis attributable to high BMI across 204 countries, 21 GBD regions, and 5 Socio-Demographic Index (SDI) tiers. Temporal trends in ASDR were quantified using the estimated annual percentage change (EAPC) and Joinpoint regression. Age-period-cohort models and decomposition analysis identified drivers of burden, while inequality was assessed using the Slope Index of Inequality (SII) and Concentration Index (CI). ARIMA models projected trends to 2036; 2) restructuring results to emphasize major findings (e.g., ASDR trends, regional differences, and projections). These changes reduce word count while maintaining scientific rigor, ensuring clarity for readers.

2.Comment: The graphs and tables are well-designed; however, for certain countries with negative confidence intervals, the authors may consider commenting on the statistical accuracy of those estimates.

Response: We appreciate the reviewers' concern about the statistical accuracy of our estimates. The negative confidence interval for ASDR stems from the Bayesian modeling framework used in the GBD study, which utilizes prior distributions to infer data from regions with limited epidemiological records. Negative confidence intervals are a product of the characteristics of the statistical model and data limitations, and do not reflect the direction of the true burden. We have added this section to the discussion (Discussion of limitations on page 36).

3.Comment: The reference section could be further strengthened by including more recent sources from 2022–2024, although the current references are valid and relevant.

Response: Thank you for the valuable suggestion to strengthen the reference section with more recent sources (2022–2024). We have revised references 3, 4, 10, 11, 21, 42, and 43 and updated them as follows:

[3] Hallberg S, Rolfson O, Karppinen J, et al. Economic burden of osteoarthritis - multi-country estimates of direct and indirect costs from the BISCUITS study. Scand J Pain. 2023;23 (4):694-704. doi:10.1515/sjpain-2023-0015

[4] Xiang L, Graves N, Low AHL, et al. Cost of lost productivity in inflammatory arthritis and osteoarthritis in the year before and after diagnosis: An inception cohort study. Int J Rheum Dis. 2024;27 (7):e15252. doi:10.1111/1756-185X.15252

[10] Batushansky A, Zhu S, Komaravolu RK, et al. Fundamentals of OA. An initiative of Osteoarthritis and Cartilage. Obesity and metabolic factors in OA. Osteoarthritis Cartilage. 2022;30 (4):501-515. doi:10.1016/j.joca.2021.06.013

[11] Binvignat M, Sellam J, Berenbaum F, et al. The role of obesity and adipose tissue dysfunction in osteoarthritis pain. Nat Rev Rheumatol. 2024;20 (9):565-584. doi:10.1038/s41584-024-01143-3

[21] Global Burden of Disease Collaborative Network. Global Burden of Disease Study 2021 (GBD 2021) Socio-Demographic Index (SDI) 1950–2021. Seattle, United States of America: Institute for Health Metrics and Evaluation (IHME), 2024

[42] Cao F, Li DP, Wu GC, et al. Global, regional and national temporal trends in prevalence for musculoskeletal disorders in women of childbearing age, 1990-2019: an age-period-cohort analysis based on the Global Burden of Disease Study 2019. Ann Rheum Dis. 2024;83 (1):121-132. doi:10.1136/ard-2023-224530

[43] Hernandez PA, Bradford JC, Brahmachary P, et al. Unraveling sex-specific risks of knee osteoarthritis before menopause: Do sex differences start early in life?. Osteoarthritis Cartilage. 2024;32 (9):1032-1044. doi:10.1016/j.joca.2024.04.015

Responses to Reviewer #3

Manuscript Code: PONE-D-25-07975

The global, regional, and national burden of early-onset osteoarthritis attributable to high BMI from 1990 to 2021, along with projections for 2036: A systematic analysis of the 2021 Global Burden of Disease Study

Comment: Thank You for submitting your work. The methodology is rigourous and the topic is interesting. The forecast modelling adds an interesting aspect to the study. Excellent work on the manuscript overall.

Response: We are sincerely grateful for your generous feedback and enthusiastic endorsement of our manuscript.

Final Remarks

We appreciate the reviewers' insightful feedback, which has significantly improved the clarity and quality of our manuscript. We have addressed all concerns and made the necessary revisions to enhance readability, formatting, and consistency.

We look forward to your further consideration of our revised manuscript. Please let us know if any additional modifications are required.

Best regards,

Kewen Li

Qinghai University Affiliated Hospital

---

## [Decision Letter · Decision Letter 1]

Global, regional, and national burden of early-onset OA attributable to high BMI: 1990–2021 estimates and 2036 projections from the global burden of disease study

PONE-D-25-07975R1

Dear Dr. Li,

We’re pleased to inform you that your manuscript has been judged scientifically suitable for publication and will be formally accepted for publication once it meets all outstanding technical requirements.

Kind regards,

Qian Wu

Academic Editor

PLOS ONE

Additional Editor Comments (optional):

Reviewers' comments:

Reviewer's Responses to Questions

**Comments to the Author**

1. If the authors have adequately addressed your comments raised in a previous round of review and you feel that this manuscript is now acceptable for publication, you may indicate that here to bypass the “Comments to the Author” section, enter your conflict of interest statement in the “Confidential to Editor” section, and submit your "Accept" recommendation.

Reviewer #1: All comments have been addressed

2. Is the manuscript technically sound, and do the data support the conclusions?

Reviewer #1: Yes

3. Has the statistical analysis been performed appropriately and rigorously? 

Reviewer #1: Yes

4. Have the authors made all data underlying the findings in their manuscript fully available?

Reviewer #1: Yes

5. Is the manuscript presented in an intelligible fashion and written in standard English?

Reviewer #1: Yes

6. Review Comments to the Author

Reviewer #1: (No Response)

7. PLOS authors have the option to publish the peer review history of their article (what does this mean? ). If published, this will include your full peer review and any attached files.

**Do you want your identity to be public for this peer review?** For information about this choice, including consent withdrawal, please see our Privacy Policy .

Reviewer #1: **Yes: ** Dr Satabdi Mitra

---

## [Editor Report · Acceptance letter]

PONE-D-25-07975R1

PLOS ONE

Dear Dr. Li,

I'm pleased to inform you that your manuscript has been deemed suitable for publication in PLOS ONE. Congratulations! Your manuscript is now being handed over to our production team.

Kind regards,

on behalf of

Dr. Qian Wu

Academic Editor

PLOS ONE